# Learning to Stop: Deep Learning for Mean Field Optimal Stopping

**Lorenzo Magnino** [* 1]   **Yuchen Zhu** [* 2]   **Mathieu Lauriere** [1 3]

## Abstract

Optimal stopping is a fundamental problem in optimization with applications in risk management, finance, robotics, and machine learning. We extend the standard framework to a multi-agent setting, named multi-agent optimal stopping (MAOS), where agents cooperate to make optimal stopping decisions in a finite-space, discrete-time environment. Since solving MAOS becomes computationally prohibitive as the number of agents is very large, we study the mean-field optimal stopping (MFOS) problem, obtained as the number of agents tends to infinity. We establish that MFOS provides a good approximation to MAOS and prove a dynamic programming principle (DPP) based on mean-field control theory. We then propose two deep learning approaches: one that learns optimal stopping decisions by simulating full trajectories and another that leverages the DPP to compute the value function and to learn the optimal stopping rule using backward induction. Both methods train neural networks to approximate optimal stopping policies. We demonstrate the effectiveness and the scalability of our work through numerical experiments on 6 different problems in spatial dimension up to 300. To the best of our knowledge, this is the first work to formalize and computationally solve MFOS in discrete time and finite space, opening new directions for scalable MAOS methods. Code is available at Learning-to-Stop.

## 1. Introduction

Optimal stopping (OS) has emerged as a powerful framework for addressing real-world problems involving uncertainty and sequential decision-making, where the goal is to determine the best time to stop a stochastic process to achieve a specific objective (see (Shiryaev, 2007), (Ekren et al., 2014) for theoretical foundations; (Lippman & McCall, 1976) for the well-known secretary problem; and (Wang et al., 1993; Dai et al., 2019) for machine learning perspectives)

The OS framework has been extended to cover multi-agent scenarios, in which the aim is to stop several (possibly interacting) dynamical systems (agents) at different times in order to minimize a common cost function. We will refer to this setting as multi-agent optimal stopping (MAOS). It has gained significant importance in a variety of fields. In robotics, it has been applied to mission monitoring tasks, where multiple robots track the progress of other robots performing a specific operation (Best et al., 2018). In finance, the problem of pricing options with multiple stopping times (see (Kobylanski et al., 2011)) can be formulated as an MAOS problem, motivating recent studies (Talbi et al., 2023; 2024).

**However, the problem's complexity increases drastically as the number of agents grows.** To address this challenge, we consider the limit as the number of agents tends to infinity and study mean-field approximations (see Fig. 1). In multi-agent control, this approach leads to the theory of Mean Field Control (MFC), which models large systems of interacting agents that cooperatively minimize a social cost by selecting optimal controls (see (Bensoussan et al., 2013; Carmona & Delarue, 2018)). Applications include crowd motion (Achdou & Laurière, 2016b; Achdou & Lasry, 2019), flocking (Fornasier & Solombrino, 2014), finance (Carmona & Laurière, 2023), opinion dynamics (Liang & Wang, 2019), and artificial collective behavior (Carmona et al., 2019; Gu et al., 2021; Cui et al., 2024). Computational methods for MFC problems include numerical methods for partial differential equations (Achdou & Laurière, 2015; Achdou & Laurière, 2016a; Briceño Arias et al., 2018; Reisinger et al., 2024), numerical methods for backward stochastic differential equations (Chassagneux et al., 2019; Balata et al., 2019), deep learning methods (Fouque & Zhang, 2020; Carmona & Laurière, 2021; Germain et al., 2022a; Dayanıklı et al., 2024) and reinforcement learning methods (Carmona et al., 2019; Gu et al., 2023; Chen et al., 2020; Carmona et al., 2023; Cui et al., 2024).

---

[*]Equal contribution, random order [1]Shanghai Frontiers Science Center of Artificial Intelligence and Deep Learning, NYU Shanghai [2]Machine Learning Center, Georgia Institute of Technology [3]NYU-ECNU Institute of Mathematical Sciences, NYU Shanghai. Correspondence to: Mathieu Lauriere <mathieu.lauriere@nyu.edu>.

*Proceedings of the 42$^{nd}$ International Conference on Machine Learning*, Vancouver, Canada. PMLR 267, 2025. Copyright 2025 by the author(s).

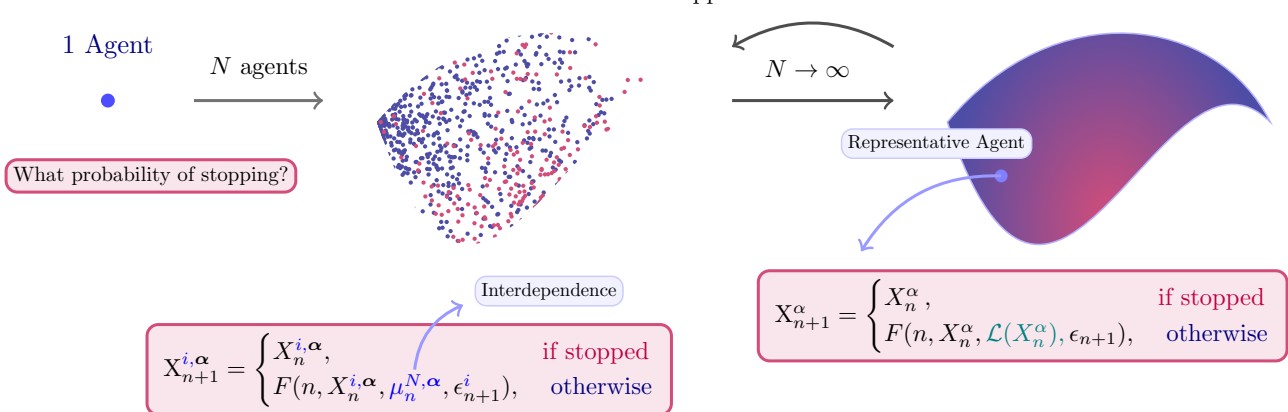

Figure 1: **Finite-agent vs mean-field:** Each agent decides to stop or continue. In the multi-agent setting, agents interact via the empirical distribution. As $N \to \infty$, this interaction is captured by the Ap. The approximation theorem (Thm 3.2) ensures that MFOS is a good proxy for MAOS.

In contrast with optimal control and Markov decision processes, mean-field approximations have not been used for OS problems, except for (Talbi et al., 2024; 2023; Yu & Yuan, 2023; Agram & Øksendal, 2024; He, 2025) in the continuous time and continuous space setting, and **computational methods have not yet been developed**.

Our work takes a first step in this direction by focusing on discrete-time, finite-space MFOS models. We establish a theoretical foundation and introduce two deep learning methods capable of solving MFOS with many states by learning optimal stopping decisions as functions of the entire population distribution. We refer to these methods as the Direct Approach (DA) and the Dynamic Programming Approach (DP) and evaluate their performance across multiple environments (see Fig. 2).

> **Contributions.** Our main contributions are twofold:
>
> *Theoretically*: we provide a DPP for MFOS and we prove that MFOS in discrete space and time yields to an approximate optimal stopping decision for $N$-agent MAOS with a rate of $\mathcal{O}(1/\sqrt{N})$ (Thm 3.2).
>
> *Computationally*: we propose two deep learning methods to solve MFOS problems, by learning the optimal stopping decision as a function of the whole population distribution (Alg. 2 and 1).

To the best of our knowledge, this is the first work to study discrete-time, finite-space MFOS problems. Our theoretical results rely on the interpretation of MFOS problems as MFC problems, which provides a new perspective and opens up new directions to study MFOS problems. Additionally, it is the first time that computational methods are proposed to solve MFOS. This is a first step towards solving complex

MAOS problems with a large number of agents.

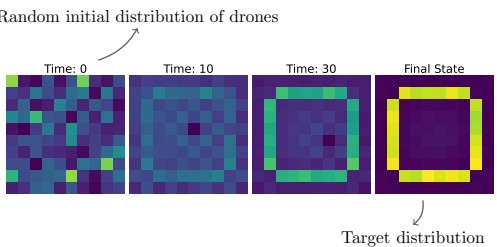

Figure 2: An infinite population of agents (e.g., drones) evolves from a random initial distribution to a target one by stopping each agent at the right time (Appx E.6).

**Related works.** MFOS has been recently studied in continuous time and space from a purely theoretical view by Talbi et al. (2023; 2024) who studied the connection with finite-agent MAOS problems and characterized the solution of (continuous) MFOS using a PDE on the infinite-dimensional space of probability measures, which is intractable. Instead, we focus on *discrete time* scenarios with *finite state space* (i.e., an individual agent's state can take only finitely many different values), and hence the distribution is finite dimensional. This setting can be viewed as an approximation of the continuous setting. Another difference with (Talbi et al., 2023; 2024) is that these work purely rely on an OS viewpoint, while we unveil a connection with MFC problems. This is a conceptual contribution of our work. Some popular classical methods for OS problems are the least-square Monte Carlo approach of (Longstaff & Schwartz, 2001) for American options in finance and numerical methods for partial differential equations (Achdou & Pironneau, 2005). Early works on machine learning methods for optimal stopping problems include the dynamic programming

approach of (Tsitsiklis & Van Roy, 1999). Deep learning methods have been proposed for discrete time *single-agent* OS problems. (Becker et al., 2019) proposed to learn the stopping decision at each time using a deep neural network. (Herrera et al., 2023) extended the approach using randomized neural networks. (Damera Venkata & Bhattacharyya, 2024) proposed to use recurrent neural networks to solve non-Markovian OS problems. Other approaches have been proposed, particularly for continuous-time OS problems, such as learning the stopping boundary (Reppen et al., 2022). These single-agent OS approaches cannot be easily adapted to solve MAOS problems: the solution obtained by treating the whole system as one agent would lead to stopping all the agents at the same time, and, more importantly, single-agent methods do not capture the interdependence between agents. Furthermore, these approaches are not suitable to tackle continuous space MFOS problems as introduced by Talbi et al. (2023) because the value function must be a function of the population distribution, which leads to an infinite-dimensional problem. For this reason, there are *no existing computational methods for MFOS*.

## 2. Model

When the number of agents tends to infinity, an aggregation effect takes place, allowing us to represent the influence of the community using an "average" term, commonly referred to as the *mean-field* term. As the number of agents approaches infinity, they become independent and identically distributed (i.i.d.), and the behavior of each individual agent is determined by a stochastic differential equation (SDE) of McKean-Vlasov type. This phenomenon is often known as the "propagation of chaos" (Sznitman, 1991). The objective is to discern the properties of the solutions to the limiting problem. By integrating these properties into the formulation of the $N$-agent control framework, we can derive approximate solutions to the latter problem (for more theoretical background on MFC, see (Bensoussan et al., 2013; Carmona et al., 2013; Carmona & Delarue, 2018)).

### 2.1. Motivation and challenges: finite agent model

The mean-field problem that we will solve is motivated by the $N$-agent problem that we are about to describe. Let $\mathcal{X}$ be a finite state space. Let us denote by $\mathcal{P}(\mathcal{X})$ the set of probability distributions on $\mathcal{X}$, and let $E$ be the set of realizations of the random noise. Let $T$ be a time horizon and let $N$ be the number of agents that are interacting.. Each agent $i$ has a state denoted by $X_n^i$ at time $n$. At time $n$, each agent stops with probability $p_n^i(\boldsymbol{X}_n^{\boldsymbol{\alpha}})$, where $p_T^i(\cdot) = 1$. We introduce $\alpha_n^i$ a random variable taking value 0 if the agent continues and 1 if it stops. We denote by $\pi_n^i(\cdot | \boldsymbol{X}_n^{\boldsymbol{\alpha}}) = Be(p_n^i(\boldsymbol{X}_n^{\boldsymbol{\alpha}}))$ its distribution, which is a Bernoulli distribution. We denote by $\boldsymbol{X}_n^{\alpha} = (X_n^1, \ldots, X_n^N)$

and $\boldsymbol{\alpha} = (\alpha^1, \ldots, \alpha^N)$ the vectors of states and stopping decisions at time $n$.

**Dynamics.** We assume that the agents are indistinguishable and interact in a symmetric fashion, i.e. through their empirical distribution $\mu_n^{N,\boldsymbol{\alpha}}(x) := \frac{1}{N} \sum_{i=1}^N \delta_{X_n^{i,\alpha}}(x)$, which is the proportion of agents at $x$ at time $n$ with $\delta$ the indicator function. The system evolves according to a transition function $F : \mathbb{N} \times \mathcal{X} \times \mathcal{P}(\mathcal{X}) \times E \to \mathcal{X}$. In particular: for every $i = 1, \ldots, N$,

$$\begin{cases} X_0^{i,\boldsymbol{\alpha}} \sim \mu_0 \qquad \alpha_n^i \sim \pi_n^i(\cdot | \boldsymbol{X}_n^{\boldsymbol{\alpha}}), \\ X_{n+1}^{i,\boldsymbol{\alpha}} = \begin{cases} F(n, X_n^{i,\boldsymbol{\alpha}}, \mu_n^{N,\boldsymbol{\alpha}}, \epsilon_{n+1}^i), & \text{if } \sum_{m=0}^n \alpha_m^i = 0 \\ X_n^{i,\boldsymbol{\alpha}}, & \text{otherwise,} \end{cases} \end{cases}$$

(1)

where $\epsilon_n^i$ is a random noise that affects the evolution of agent $i$ and $m_0$ is the initial distribution.

Let us define the stopping time for agent $i$: $\tau^i = \inf\{n \geq 0 : \sum_{m=0}^n \alpha_m^i \geq 1\}$, which is the first time for player $i$ that the decision is to stop. Note that, since $p_T^i(\cdot) = 1$, this means that $\tau^i \leq T$ for all $i = 0, \ldots, N$.

**Objective function.** Let us consider a function $\Phi : \mathcal{X} \times \mathcal{P}(\mathcal{X}) \to \mathbb{R}$. $\Phi(x, \mu)$ denotes the cost that an agent incurs if she stops at $x$ and the current population distribution is $\mu$. The goal for all the $N$ agents is to collectively minimize the following social cost function:

$$J^N(\alpha^1, \ldots, \alpha^N) = \mathbb{E}\left[\frac{1}{N} \sum_{i=1}^N \Phi(X_{\tau^i}^{i,\boldsymbol{\alpha}}, \mu_{\tau^i}^{N,\boldsymbol{\alpha}})\right]. \quad (2)$$

The problem consists in finding the best controls $(\alpha^1, \ldots, \alpha^N) \in \arg\min J^N$. Next, we give an example.

**Motivating Example:** We take as a state space $\mathcal{X} = \{1, 2, 3, 4, 5, 6, 7\}$ with boundaries (i.e., in 1 agents cannot move left and in 7 they cannot move right), time horizon $T = 3$, transition function $F(n, x, \mu, \epsilon) = x + \epsilon$, where $\epsilon = 0$ with probability $p = 1/2$, $\epsilon = 1$ with probability $p = 1/4$ and $\epsilon = -1$ with probability $p = 1/4$. Following (1), the dynamics of agent $i$ is: $X_{n+1}^i = X_n^i + \epsilon_{n+1}^i$ if the agent does not stop, and $X_{n+1}^i = X_n^i$ otherwise. All agents start in $x = 4$. We define a target distribution $\rho_{\text{target}} = \frac{1}{2}\delta_4 + \frac{1}{4}\delta_5 + \frac{1}{4}\delta_3$. If the agent $i$ stops at time $n$, then she is incurred the cost: $\Phi(X_n^i, \mu_n^N) = \sum_{x \in \mathcal{X}} |\mu_n^N(x) - \rho_{\text{target}}(x)|^2$, which is smaller if the agent stops when the population distribution matches the target one. **Notice that some agents might have to stop even though the target distribution is not matched, so that other agents can later have a lower cost because this is a *cooperative* task.** Solving exactly this problem (i.e., finding the optimal stopping time for every agent) is very complex. Our approach is to consider the mean-field problem, which leads to an efficient approximate solution (see Example 4 in Section 6).

**Challenges:** Single-agent methods cannot be readily applied to the multi-agent setting since they cannot capture the interdependence due to the distribution in the cost and in the dynamics. In particular, in the multi-agent setting, we allow agents to stop at different times. When the number of agents is very large, computing exactly the optimal stopping times is infeasible. Mean-field optimal stopping (MFOS) can intuitively provide an approximate solution, but (1) this needs to be justified, and (2) scalable numerical methods for MFOS need to be developed.

## 2.2. Mean-field model

As mentioned earlier, if we let the number of players tend to infinity, we expect, thanks to propagation of chaos type results, that the states will become independent and each state will have the same evolution, which will be a nonlinear Markov chain. More precisely, passing formally to the limit in the dynamics (1), we obtain the following evolution:

$$
\begin{cases}
X_0^\alpha \sim \mu_0 \qquad \alpha_n \sim \pi_n(\cdot|X_n^\alpha) = Be(p_n(X_n^\alpha)) \\
X_{n+1}^\alpha = \begin{cases} F(n, X_n^\alpha, \mu_n^\alpha, \epsilon_{n+1}), & \text{if } \sum_{m=0}^{n} \alpha_m = 0 \\ X_n^\alpha, & \text{otherwise,} \end{cases}
\end{cases}
$$
(3)

where $p_n(x)$ denotes the probability with which the agent continues if she is in state $x$ at time $n$, and $\mu_n^\alpha$ is the distribution of $X_n^\alpha$ itself, which we may also denote by $\mathcal{L}(X_n^\alpha)$. We want to emphasize the fact that the introduction of randomized stopping times for individual agents is crucial for our purpose; see the example in Appx. A.1.

We can define, in the same way we did before, the first time at which the control $\alpha$ is 1 as $\tau := \inf\{n \geq 0 : \sum_{m=0}^{n} \alpha_m \geq 1\}$. Then the social cost function in the mean-field problem is defined as:

$$
J(\alpha) = \mathbb{E}\left[\Phi(X_\tau^\alpha, \mathcal{L}(X_\tau^\alpha))\right].
$$
(4)

Notice that here the expectation has the effect of averaging over the whole population, so there is no counterpart to the empirical average that appears in the finite agent cost (2). To stress the dependence on the initial distribution, we will sometimes write $J(\alpha, m_0)$.

## 2.3. Mean-field model with extended state

A key step towards building efficient algorithms is dynamic programming, which relies on the Markovian property. However, in its current form, the above problem is not Markovian. To ensure Markovianity, we need to keep track of the information about whether the player's process has been stopped in the past. This information is not contained in the state, so we need to extend the state. Let

$A^\alpha = (A_n^\alpha)_{n=0,\ldots,T}$ the process such that $A_n^\alpha = 0$ if the agent has *already* stopped before time $n$, and 1 otherwise. We can interpret this process as the "Alive" process, while $\alpha$ stands for the "action", namely, to stop or not. So $A_n^\alpha = 1$ means the agent has not stopped yet; when the agent stops, $\alpha_n = 1$ and $A_{n+1}^\alpha$ switches to 0. It is important to notice that if the agent is stopped precisely at time $n$ then, we still have $A_n^\alpha = 1$ but $A_m^\alpha = 0$ for every $m > n$. We define the *extended state* as: $Y_n^\alpha = (X_n^\alpha, A_n^\alpha)$, which takes value in the extended state space $\mathcal{S} := \mathcal{X} \times \{0, 1\}$. Then, the dynamics (3) of the representative player can rewritten as:

$$
\begin{cases}
X_0^\alpha \sim \mu_0, \qquad A_0^\alpha = 1 \\
\alpha_n \sim \pi_n(\cdot|X_n^\alpha) = Be(p_n(X_n^\alpha)); \ A_{n+1}^\alpha = A_n^\alpha(1 - \alpha_n) \\
X_{n+1}^\alpha = \begin{cases} F(n, X_n^\alpha, \mathcal{L}(X_n^\alpha), \epsilon_{n+1}), \text{if } A_n^\alpha(1 - \alpha_n) = 1 \\ X_n^\alpha, \qquad \text{otherwise.} \end{cases}
\end{cases}
$$
(5)

The idea of extending the state using the extra information is similar to Talbi et al. (2023) in continuous time and space. The mean-field social cost (4) can rewritten as:

$$
J(\alpha) = \mathbb{E}\left[\sum_{m=0}^{T} \Phi(X_m^\alpha, \mathcal{L}(X_m^\alpha))A_m^\alpha \alpha_m\right]
$$
(6)

Actually, notice that the expectation amounts to taking a sum with respect to the extended state's distribution. Let us denote by $\nu_n^p = \mathcal{L}(Y_n^\alpha)$ the distribution at time $n$. We denote $\nu_X^p$ the first marginal of $\nu^p$ (sometimes also denoted by $\mu$). Note that it does not depend on $\alpha$ but only on the stopping probability $p$, so we use the superscript $p$ when referring to $\nu$. This distribution evolves according to the mean-field dynamics:

$$
\begin{cases}
\nu_0^p(x, 0) = 0, \quad \nu_0^p(x, 1) = \mu_0(x), \qquad x \in \mathcal{X}, \\
\nu_{n+1}^p = \bar{F}(\nu_n^p, p_n),
\end{cases}
$$
(7)

where the function $\bar{F}$ is defined as follows. We denote by $\mathcal{H}$ the set of all functions $h : \mathcal{X} \to [0, 1]$, which represent a stopping probability (for each state). Then, $\bar{F} : \{0, \ldots, T\} \times \mathcal{P}(\mathcal{S}) \times \mathcal{H} \to \mathcal{P}(\mathcal{S})$ is defined, for every $x \in \mathcal{X}, a \in \{0, 1\}$, by:

$$
(\bar{F}(\nu, h))(x, a) = \left(\nu(x, 0) + \nu(x, 1)h(x)\right)(1 - a) +
$$
$$
\left(\sum_{z \in \mathcal{X}} \nu(z, 1)\left(q_{z,x}^\nu(1 - h(z))\right)\right)a, \quad (8)
$$

where $\mathcal{Q}^\nu = (q_{z,x}^\nu)_{z,x \in \mathcal{X}}$ is the transition matrix associated to the unstopped process $X$, i.e. $q_{z,x}^\nu$ is the probability to go from the state $z$ to the state $x$ knowing that we are not going to stop in $x$. Notice that in general the transitions may depend on $\nu$ itself, which is why this type of dynamics

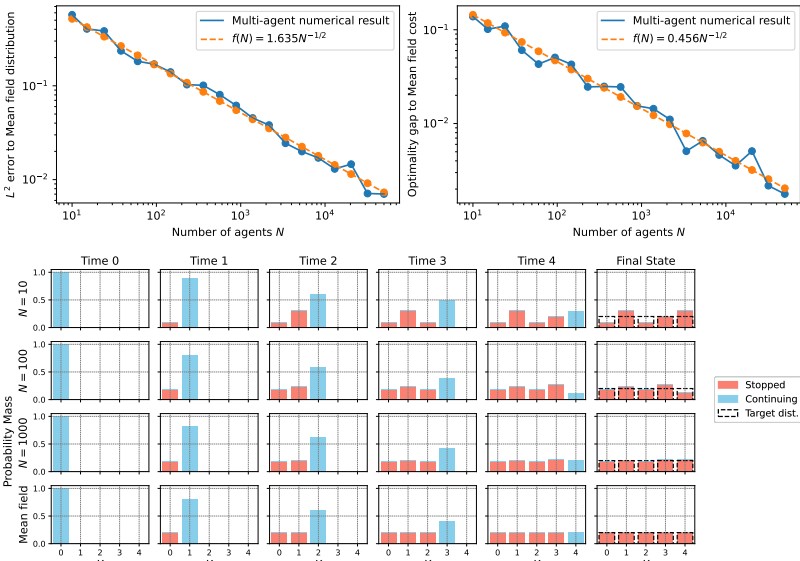

Figure 3: MFOS v.s. MAOS. We use the stopping probability function learned by Algorithm 1 for MFOS to simulate the multi-agent OS. **Top left:** $L^2$ distance of multi-agent empirical distribution to mean-field distribution, averaged over 10 runs. **Top right:** Optimality gap between multi-agent and mean-field cost, averaged over 10 runs. **Bottom:** Evolution of multi-agent empirical distribution for different agents $N$.

is sometimes referred to *nonlinear* dynamics. The mean-field social cost can be rewritten purely in terms of the distribution as follows:

$$J(p) = \sum_{m=0}^{T} \sum_{(x,a) \in \mathcal{S}} \nu_m^p(x,a)\Phi(x,\mu_m^p)ap_m(x), \quad (9)$$

where $p : \{0, \ldots, T\} \times \mathcal{X} \to [0,1]$ is the function that associates to every time step and state the probability to stop (in that state at that time). Let us define $\mathcal{P}_{0,T}$ to be the set of all such functions.

The link with the formulation in (6) is that $\alpha_n(x)$ is distributed according to $Be(p_m(x))$, and $\nu_m^p := \mathcal{L}(Y_m^\alpha)$ is the extended state's distribution. Moreover, $\nu_m^p(x,0)$ is the mass in $x$ that has stopped. Last, $\mathcal{L}(X_m^\alpha) = \mu_m^\alpha(x) = \sum_{a \in \{0,1\}} \nu_m^p(x,a)$ is the first marginal of this distribution.

## 3. Approximate optimality for finite-agent model

In this section, we aim to address the following questions:
**Q:** *"Is the mean-field model capable of solving the original problem of $N$ agents? If so, in what sense?"*

Specifically, we demonstrate that the MFOS solution provides an approximately optimal solution for the finite-agent MAOS problem. The main assumption we use is:

**Assumption 3.1.** Let $L_p > 0$ and let us define $\mathcal{P} := \{p : \{0, \ldots, T\} \times \mathcal{X} \times \mathcal{P}(\mathcal{S}) \to [0,1] : p$ is $L_p$-Lipschitz$\}$, the set of all possible admissible policies $p$. Assume that the *mean-field* dynamics $\bar{F}$ described in (8) is $L_{\bar{F}}$ - Lipschitz.

Assume also that the function $\Psi : \mathcal{P}(\mathcal{X} \times \{0,1\}) \times \mathcal{P}(\mathcal{X}) \to \mathbb{R}$ defined as $\Psi(\nu, h) := \sum_{(x,a) \in \mathcal{S}} \nu(x,a)\Phi(x, \nu_X)ah(x)$ is $L_\Psi$-Lipschitz.

Assuming Lipschitz dynamics, cost, and policies is classical in the literature on mean-field control problems, see e.g. (Mondal et al., 2022; Pásztor et al., 2023; Cui et al., 2023) and can be achieved using neural networks (Araujo et al., 2022). Due to space constraints, we simply provide an informal statement here. The precise statement is deferred to Appx. A.2, see Theorem A.3, along with the detailed setting and notations.

**Theorem 3.2** ($\varepsilon$-approximation of the $N$-agent problem). *Suppose Assumption 3.1 holds. If $p^*$ is the optimal policy for the MFOS problem and $\hat{p}$ is the optimal policy for the $N$-agent problem (when all the agents use the same policy), then: as $N \to +\infty$, $J^N(p^*, \ldots, p^*) - J^N(\hat{p}, \ldots, \hat{p}) \to 0$, with rate of convergence $\mathcal{O}\left(1/\sqrt{N}\right)$ (the explicit bound is given in the proof).*

A key step in the proof consists in analyzing the difference between the $N$-agent dynamics and the mean-field dynamics under a stopping policy, see Lemma A.1 ("Convergence of the measure") in the Appendix.

Theorem 3.2 is further supported through empirical evidence as is shown in Fig. 3, where we apply the stopping probability function learned by Algorithm 2 on MFOS in Example 1 (see Section 5 and 6 for details) to the $N$-agent problem with varying $N$ (see Appx. E.1). We compute the $L^2$ distance of

multi-agent empirical distribution to mean-field distribution and the optimality gap between multi-agent and mean-field cost, both averaged over 10 runs. The plots demonstrate a clear decay rate of order $N^{-1/2}$. *This theorem justifies that MFOS is not only an intrinsically interesting problem, but the solution to MFOS also serves as an approximate solution to the corresponding MAOS problem.* In the sequel, we will focus on solving the MFOS problem.

## 4. Dynamic programming

Our motivation for developing a dynamic programming principle (DPP) for our formulation comes from both the literature and numerical purposes. In the control theory of a dynamic system, it has been studied and used very often to find solutions to a given optimization problem. Moreover, implementing an algorithm that is built on DPP often leads to precise optimal solutions that perform better than other methods.

We introduce the dynamical form of the social cost (9) as:

$$V_n(\nu) := \inf_{p \in \mathcal{P}_{n,T}} J(p(x), \nu) \tag{10}$$

$$:= \inf_{p \in \mathcal{P}_{n,T}} \sum_{m=n}^{T} \sum_{(x,a) \in \mathcal{S}} \nu_m^{p,\nu,n}(x,a) \Phi(x, \mu_m^{p,\nu,n}) a p_m(x),$$

where $\mathcal{P}_{n,T}$ is the set of all possible function $p : \{n, \ldots, T\} \times \mathcal{X} \to [0,1]$ and $\nu^{p,\nu,n}$ denotes the distribution of the process that starts at time $n$ with a given distribution $\nu$; it satisfies (7) but starting at time $n$ instead of 0 with $\nu_n^{p,\nu,n} = \nu$. The optimal value at time 0 will be denoted: $V^*(\nu) = V_0(\nu)$, which is also equal to $\inf_p J(p, \nu)$. We can now state and prove the following DPP.

**Theorem 4.1** (Dynamic Programming Principle). *For the dynamics given by (5) and the value function given by (10) the following dynamic programming principle holds:*

$$\begin{cases} V_T(\nu) = \sum_{(x,a) \in \mathcal{S}} \nu(x,a) \Phi(x, \nu_X) a, \\ V_n(\nu) = \inf_{h \in \mathcal{H}} \sum_{(x,a) \in \mathcal{S}} \nu(x,a) \Phi(x, \nu_X) a h(x) \\ \qquad + V_{n+1}(\bar{F}(\nu, h)), \qquad n < T, \end{cases} \tag{11}$$

*where $\nu_X$ is the first marginal of the distribution $\nu$, i.e., $\nu_X(x) = \nu(x,0) + \nu(x,1)$. The sequence of optimizers define an optimal stopping decision that we will denote by $h^* : \{0, \ldots, T-1\} \times \mathcal{X} \times \mathcal{P}(\mathcal{S}) \to [0,1]$ and satisfies: for every $n \in \{0, \ldots, T-1\}$ and every $\nu \in \mathcal{P}(\mathcal{S})$, $V_n(\nu) = \sum_{(x,a) \in \mathcal{S}} \nu(x,a) \Phi(x, \nu_X) a h_n^*(x, \nu) + V_{n+1}(\bar{F}(\nu, h_n^*(x, \nu)))$.*

To prove this result, we will show that we can reduce the problem to a mean-field optimal control problem in discrete time and continuous space. See details in Appx. B.

Dynamic programming for MFC problem (Laurière & Pironneau, 2014; Pham & Wei, 2017) and mean-field MDPs (Gu et al., 2023; Motte & Pham, 2022; Carmona et al., 2023) have been extensively studied, and DPP for continuous time MFOS has been established by (Talbi et al., 2023) using a PDE approach. However, to the best of our knowledge, this is the first DPP result for MFOS problems in discrete time. It serves as a building block for one of the deep learning methods proposed below.

---
**Algorithm 1** Dynamic Programming (DP)
---
**Require:** stopping decision neural networks: $\psi_\theta^n : \mathcal{X} \times \mathcal{P}(\mathcal{S}) \to [0,1]$ for $n \in \{0, \ldots, T-1\}$, max training iteration $N_{\texttt{iter}}$.
1: Set $\psi_\theta^T = 1$
2: **for** $n = T-1, \ldots, 0$ **do**
3:      **for** $k = 0, \ldots, N_{\texttt{iter}} - 1$ **do**
4:          Sample $\nu_n^p$
5:          **for** $m = n, \ldots, T$ **do**
6:              **if** $m = n$ **then**
7:                  $p_m(x) = \psi_\theta^m(x, \nu_m^p; \theta_k^n)$
8:              **else**
9:                  $p_m(x) = \psi_\theta^m(x, \nu_m^p; \theta^{m,*})$
10:              $\ell_m = \sum_x \nu_m^p(x,1) \Phi(x, \mu_m) p_m(x)$
11:              $\nu_{m+1}^p = \bar{F}(\nu_m^p, p_m)$
12:          $\ell = \sum_{m=n}^{T} \ell_m$
13:          $\theta_{k+1}^n = \texttt{optim\_up}(\theta_k^m, \ell(\theta_k^n))$
14:      Set $\theta^{n,*} = \theta_{N_{\texttt{iter}}}^n$
15: **return** $(\psi_{\theta^{n,*}}^n)_{n=0,\ldots,T}$
---

In fact, we can show that this DPP still holds for a restricted class of randomized stopping times in which all the agents (regardless of their own state) have the same probability of stopping. Let $\tilde{\mathcal{P}}_{n,T}$ be the set of $p : \{0, \ldots, T\} \to [0,1]$. Notice that here $p_n$ does not depend on the individual state $x$. At every time step $n = m$, all agents have the same probability to stop $p_m$, i.e., for every $x \in \mathcal{X}$ at time $n = m$, $p_n(x) = p_n$. We call this set *synchronous* stopping times. Let us define the value:

$$\tilde{V}_n(\nu) := \inf_{p \in \tilde{\mathcal{P}}_{n,T}} \sum_{m=n}^{T} p_m \sum_{(x,a) \in \mathcal{S}} \nu_m^{p,\nu,n}(x,a) \Phi(x, \mu_m^{p,\nu,n}) a.$$

**Theorem 4.2.** *For the setting of synchronous stopping times, the value function satisfies:*

$$\begin{cases} \tilde{V}_T(\nu) = \sum_{(x,a) \in \mathcal{S}} \nu(x,a) \Phi(x, \nu_X) a, \\ \tilde{V}_n(\nu) = \inf_{h \in [0,1]} \sum_{(x,a) \in \mathcal{S}} \nu(x,a) \Phi(x, \nu_X) a h \\ \qquad + V_{n+1}(\bar{F}(\nu, h)), \qquad n < T. \end{cases} \tag{12}$$

The proof follows the same argument as that of Theorem 4.1, therefore we omit it here.

---

**Algorithm 2** Direct Approach (DA)

---

**Require:** time-dependent stopping decision neural network: $\psi_\theta : \{0, \ldots, T\} \times \mathcal{X} \times \mathcal{P}(\mathcal{S}) \to [0, 1]$, max number of training iteration $N_{\texttt{iter}}$

1: **for** $k = 0, \ldots, N_{\texttt{iter}} - 1$ **do**
2:      Sample initial $\nu_0^p$
3:      **for** $n = 0, \ldots, T$ **do**
4:          $p_n(x) = \psi_\theta(x, \nu_n^p, n; \theta_k), x \in \mathcal{X}$
5:          $\ell_n = \bar{\Phi}(\nu_n^p, p_n)$
6:          $\nu_{n+1}^p = \bar{F}(\nu_n^p, p_n)$
7:      $\ell = \sum_{n=0}^{T} \ell_n$
8:      $\theta_{k+1} = \texttt{optim\_up}(\theta_k, \ell(\theta_k))$
9: Set $\theta^* = \theta_{N_{\texttt{iter}}}$
10: **return** $\psi_{\theta^*}$

---

## 5. Algorithms

To address the MFOS problem numerically, we propose two approaches based on two different formulations. As the most naive approach, we can attempt to directly minimize the mean-field social cost $J(p)$ stated in (9), where we optimize over all the possible stopping probability functions $p : \{0, \ldots, T\} \times \mathcal{X} \to [0, 1]$. A more ideal treatment is to leverage the Dynamic Programming Principle (DPP) discussed in Theorem 4.1 and solve for the optimal stopping probability using induction backward in time. For each of the timestep $n$, we implicitly learn the true value function $V_n(\nu)$ by solving the optimization problem in (11), where we search over all possible one-step stopping probability function $h : \mathcal{X} \to [0, 1]$ for each time $n$. We refer to the method of directly optimizing mean-field social cost as the direct approach (DA) and the attempt to solve MFOS via backward induction of the DPP approach. Short versions of the pseudocodes are presented in Alg. 2 and 1. Long versions are in Appx. C (see Alg. 3 and 4). To alleviate the notations, we denote: $\bar{\Phi}(\nu, h) = \sum_{x \in \mathcal{X}} \nu(x, 1) \Phi(x, \nu_X) h(x)$, which represents the one-step mean-field cost. In the code, $\texttt{optim\_up}$ denotes one update performed by the optimizer (e.g. Adam in our simulations).

## 6. Experiments

In this section, we present 6 experiments of increasing complexity to validate our proposed method and demonstrate its potential applications. Due to space constraints, two of them have been included in Appx. E. It is important to emphasize that each experiment reflects a distinct scenario, varying both in dynamics (random, deterministic; with or without mean-field interactions) and in the cost function (with or without mean-field dependence). This provides a comprehensive overview of the method's versatility and potential applications. We solve all 6 environments with

both algorithms (the details are in Appx. E).

**Problem Dimensions:** For the problem dimension, we count it as the sum of the dimensions of the information input to the neural network. Since the state is in $\mathcal{X}$, which is finite, we encode it as a one-hot vector in $\mathbb{R}^{|\mathcal{X}|}$ before passing it to the neural network to ensure differentiability. For the mean-field distribution with stopped and non-stopped parts, it is an element of the $(2|\mathcal{X}| - 1)$-simplex, and is represented as a non-negative vector in $\mathbb{R}^{2|\mathcal{X}|}$. Therefore, MFOS tasks are of spatial dimension $|\mathcal{X}| + 2|\mathcal{X}| = 3|\mathcal{X}|$, $|\mathcal{X}|$ being the cardinality of an individual agent's state space.

**Comparison of the Two Proposed Algorithms:** While in theory both algorithms are equally capable of tackling MFOS problems, in practice, these algorithms have advantages in different settings. Empirically, we found that the optimal stopping decision is learned faster by DA than by DP when the amount of compute is not a restriction. However, DA requires differentiating through the whole trajectory at each gradient step, which requires a large amount of memory when the dimension is high. The minimum required memory for training with DA increases with $T$, whereas DP requires only constant order memory that is independent of the time horizon, since it trains independently per time step. Therefore, when targeting MFOS problems with a long time horizon $T$, DPP becomes more efficient, at least memory-wise. For similar observations in the context of continuous time optimal control, see (Germain et al., 2022b).

**Example 1** (Towards the uniform) and **Example 2** (Rolling a die), on a 1D gridworld state space, are described in details in Appx. E.1 and E.2 respectively due to space constraint.

**Example 3: Crowd Motion with Congestion.** This example extends the setting of Example 2 by incorporating a congestion term into the dynamics. The outcome of the die takes the role of the noise $\epsilon \sim \mathcal{U}(\mathcal{X})$ where $\mathcal{X} = \{1, 2, 3, 4, 5, 6\}$. The system starts in the initial distribution $\eta = \frac{1}{4}\delta_1 + \frac{1}{4}\delta_2 + \frac{1}{2}\delta_5$, and evolves according to the dynamics (5) with $\mu_0 = \eta$, and $F(n, x, \mu, \epsilon) = \epsilon$ where we are going to introduce a term of *congestion* multiplying the probability of moving by $(1 - C_{\text{cong}}\mu(x))$ to model the fact that it is difficult to move from a state $x$ if the distribution is concentrated in that state (see Appx. E.3). The social cost function associated to this scenario is $\Phi(x, \mu) = x$. Time horizon is set to $T = 4$. We perform the experiment without congestion (see Appx. E.2), and we expect congestion to slow down the movement. DA results are shown in Fig. 4.

This example demonstrates that two classes of stopping times (synchronous and asynchronous) can lead to very different optimal stopping decisions and induce distributions. Although the true value is unknown, the results indicate that synchronous stopping times yield a higher value, while asynchronous stopping times lead to a significant reduction

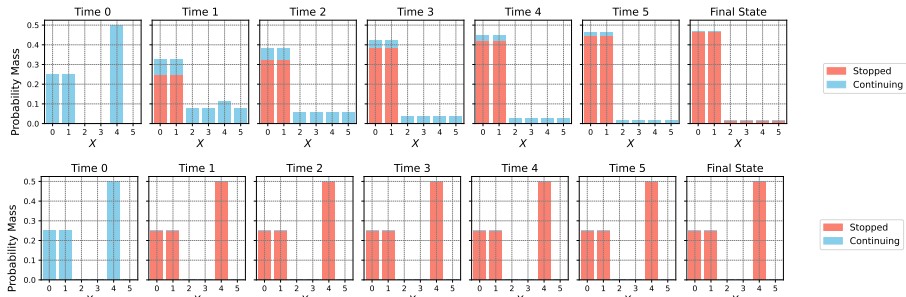

Figure 4: Example 3. DA results, asynchronous vs synchronous stopping times. Comparison of the evolution of the distribution after training (asynchronous stopping class on top, synchronous stopping class on bottom).

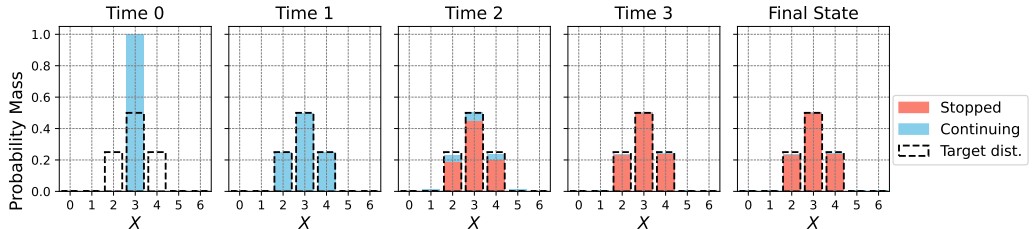

Figure 5: Example 4. DA results, asynchronous stopping. Evolution of the distribution after training.

in the cost. Additionally, in the asynchronous case, congestion leads to reduced movement, as observed between time 0 and time 1 in state 4. See Appx. E.3 for the DPP results.

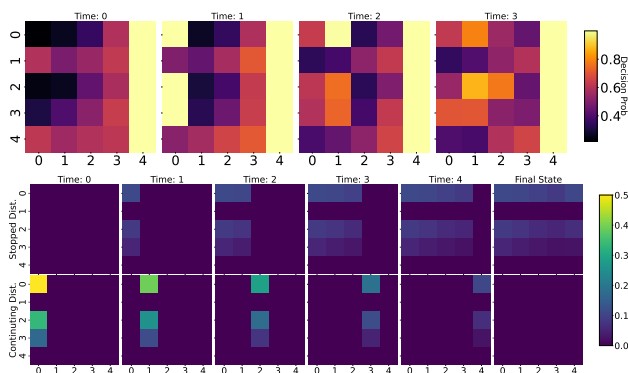

Figure 6: Example 5. DA results, asynchronous stopping. Left: stopping decision probability. Right: evolution of the distribution after training.

**Example 4: Distributional Cost.** This example extends, at the mean field level, the motivating example described at the end of Section 2.1. Based on Theorem 3.2, the mean-field solution provides a good approximation of the $N$-agent problem. DA results are shown in Fig. 5. The results for other settings are shown in Appx. E.4.

**Example 5: Towards Uniform in Dimension 2.** This example extends Example 1 (see Appx. E.1) to two dimensions, demonstrating how the algorithm performs in higher-dimensional settings. We take state space $\mathcal{X} = \{0, 1, 2, 3, 4\} \times \{0, 1, 2, 3, 4\}$, time horizon $T = 4$, transition function $F(n, x, \mu, \epsilon) = x + (1, 0)$ which means that

the agent deterministically moves to the state on the right on the same row, with boundary at $x = 4$, and cost function $\Phi(x, \mu) = \mu(x)$ which depends on the mean field only through the state of the agent (this is sometimes called local dependence). For the testing distribution, we take a distribution concentrated on state $x = 0$, denoted as $\mu_0 = \delta_0$. Fig. 6 shows that the distribution evolves towards a uniform distribution across each row, as expected, and also illustrates the optimal strategy (decision probability) required to achieve this outcome. Results for the DPP algorithm and the synchronous stopping are in Appx. E.5.

**Example 6: Matching a Target with a Fleet of Drones.** We conclude with a more realistic and complex example to showcase the potential applications of our algorithms. This example aims to align a fleet of drones with a given target distribution at terminal time $T$, starting from a random initial distribution. To make this experiment more interesting, we expand the framework described so far by considering a different type of cost and by including a noisy obstacle hindering the drones' movements (see Appx. E.6 for the mathematical formulation). We take $\mathcal{X} = \{0, \ldots, 9\} \times \{0, \ldots, 9\}$ that represents a $10 \times 10$ grid. Hence, the neural network's input is of dimension $3|\mathcal{X}| = 300$. The system follows the dynamics that diffuse uniformly over the possible neighbors, where the possible neighbors of $x \in \mathcal{X}$ are defined as $x \pm (0, 1)$ or $x \pm (1, 0)$ if the resulting state is still an element of $\mathcal{X}$. Moreover, we introduce extra stochasticity into the dynamics by placing an obstacle at a random state on the grid at each time step. The location is uniformly selected from $\mathcal{X}$ and is viewed as a *common noise* affecting the dynamics of all the agents. This introduces additional complexity in the learning problem because even

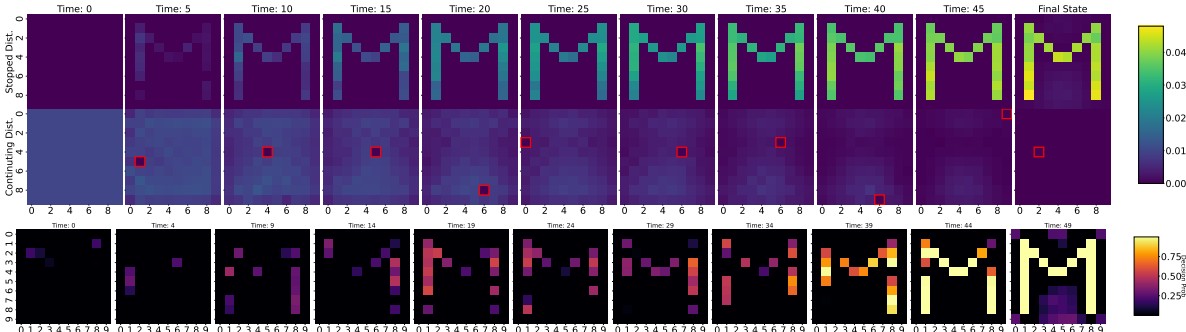

Figure 7: Example 6. DPP results, asynchronous stopping. Match the Letter "M" in $10 \times 10$ grid with common noise. We plot the stopped distribution, continuing distribution, and decision probability function every $5$ timestep. The marked red square indicates the random obstacles (common noise).

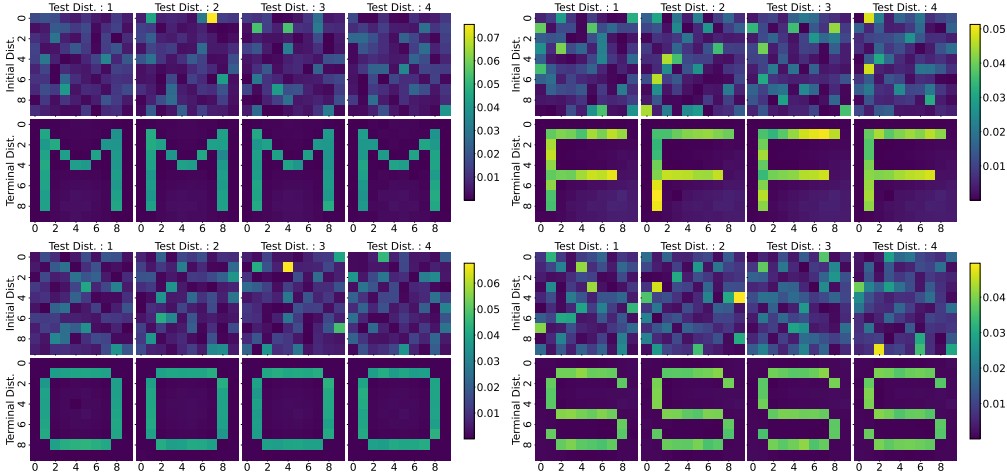

Figure 8: Example 6. DPP results, asynchronous stopping. Match the Letter "M", "F", "O", and "S". Tested with the randomly sampled initial distribution.

for a fixed stopping decision rule, the evolution of the population is stochastic. We consider the target distribution $\rho$ to be the uniform distribution over the grid of the letter "M", "F", "O", and "S" respectively, and we set the terminal cost $g_\rho(\nu) = \sum_{x \in \mathcal{X}} |\nu(x) - \rho(x)|^2$ (see results in Fig. 7). We choose the time horizon $T = 50$. Another important aspect of the algorithms' outcome is that the learned stopping decisions are *agnostic to the initial distribution* in the sense that the same stopping decision rule can be used on different initial distributions and always leads to matching the target distribution. Fig. 8 shows the terminal distributions under random initial testing distributions: the learned stopping probability function is robust to any test distribution used at inference time. Results for the DA algorithm are shown in Appx. E.6.

## 7. Conclusion

We proposed a discrete-time, finite-state MAOS problem with randomized stopping times and its mean-field version.

We proved that the latter is a good approximation of the former, and we established a DPP for MFOS. These new problems cannot be directly tackled by adapting previous methods for single-agent OS problems. To overcome these challenges, we proposed two deep learning methods and evaluated their performance over six different scenarios. When an analytical solution exists, our method recovers it in just a few iterations. In complex settings, it efficiently achieves high performance. This work lays the foundation for understanding and learning optimal stopping problems in complex interacting systems.

**Limitations and Future Works:** We did not establish a convergence proof for our algorithms due to the complexity of analyzing deep networks. Additionally, we left for future work a detailed analysis of different classes of stopping times. Finally, extending our framework to continuous spaces and validating it on real-world, ambitious applications remains an exciting direction for future research.

## Impact Statement

This paper presents work whose goal is to advance the field of Machine Learning. There are many potential societal consequences of our work, none which we feel must be specifically highlighted here.

## Acknowledgements

The authors are grateful to the anonymous reviewers and area chairs for their comments, which helped improve the quality of the paper. M. L. and L. M. were partially supported by the Shanghai Frontiers Science Center of Artificial Intelligence and Deep Learning at NYU Shanghai. M.L. was partially supported by the grant "AI-driven Initiative to Promote Research Paradigm Reform and Empower Discipline Advancement." M. L. and Y. Z. would like to thank Lexie Zhu for fruitful discussions at the early stage of this project.

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

# A. $N$-agent cooperative optimal stopping

## A.1. Why do we need randomization in the control? An Example

We want to show with an example that the extension to randomized stopping times is necessary in the mean-field formulation, because when we try to plug an optimal strategy into the $N$-agent problem, we notice that the latter is no longer optimal.

*Example* 1 (Randomized is better). Let consider the following scenario: we take the state space $\mathcal{X} = \{T, C\}$ and initial distribution $\mu_0 = 3/4\delta_T + 1/4\delta_C$; transition function $F(T, x, \mu, \epsilon) = C$, $F(C, x, \mu, \epsilon) = T$, meaning that the system at any time step, can stop or switch the state. We take as social cost:

$$\Phi(x, \mu) = \begin{cases} 1 & \text{if } \mu(x) \leq 1/2 \\ 5 & \text{if } \mu(x) > 1/2. \end{cases} \tag{13}$$

Notice that without allowing the randomized stopping the value is $V^* = 3/4 \cdot 5 + 1/4 \cdot 1 = 4$, which corresponds to stop all the distribution ( in every state) at time $n = 0$. In the end, this formulation cannot reflect the optimum in the association of $N$ agents. Indeed when we plug this policy into the $N$ agent formulation we obtained the value $V^N = 1/N(3N/4 \cdot 5 + N/4 \cdot 1) = 4$, which is not optimal since we can use the strategy ( which is going to be optimal for the $N$-agent problem) to stop, at time 0, only the $1/3$ of players in state $T$, allowing the others to change state. This leads to a final configuration of $m_1 = 1/2\delta_T + 1/2\delta_C$ and a value of $V^{*,N} = 1/N(N/4 \cdot 5 + 3N/4 \cdot 1) = 2 < V^N = 4$.

In particular, we want to emphasize the fact that, without allowing a randomized stopping time in the MF formulation, we find an optimal state-dependent strategy, which corresponds , in the problem with finite agents, to the fact that every player in the same state will have the same stopping time.

## A.2. Proof of Theorem 3.2

This section demonstrates that solving the optimal control problem at the asymptotic regime for the number of agents tending to infinity allows one to find the solution to the multi-agent problem by including the solution found at the regime in the latter. This is of fundamental importance in applications as it allows a simpler and clearer situation to be analyzed for the purpose of solving a complicated problem. Let us recall the $N$-agent formulation. We are going to work in the framework where the central planner use the same policy $p$ to control each agent. We suppose Assumption 3.1 holds.

Let us fix the following notation $\nu_m^{N,p} := \frac{1}{N}\sum_{i=1}^N \delta_{Y_m^{i,\alpha}}$ and $\nu_m^p := \mathcal{L}(Y_m^\alpha)$ .

$$\begin{cases} X_0^{i,\boldsymbol{\alpha}} \sim \mu_0, \qquad A_0^{i,\boldsymbol{\alpha}} = 1 \\ \alpha_n^i \sim \pi_n^i(\cdot|X_n^{i,\boldsymbol{\alpha}}) = Be(p_n(X_n^{i,\boldsymbol{\alpha}})) \\ A_{n+1}^{i,\boldsymbol{\alpha}} = A_n^{i,\boldsymbol{\alpha}} \cdot (1 - \alpha_n^i) \\ X_{n+1}^{i,\boldsymbol{\alpha}} = \begin{cases} F(n, X_n^{i,\boldsymbol{\alpha}}, \frac{1}{N}\sum_{j=0}^N \delta_{Y_n^{j,\alpha}}, \epsilon_{n+1}^i), & \text{if } A_n^{i,\boldsymbol{\alpha}} \cdot (1 - \alpha_n^i) = 1 \\ X_n^{i,\boldsymbol{\alpha}}, & \text{otherwise.} \end{cases} \end{cases} \tag{14}$$

The social cost is defined as:

$$\begin{aligned} J^N(p) := J^N(p, \ldots, p) &:= \frac{1}{N}\sum_{i=1}^N \mathbb{E}\left[\sum_{m=0}^T \Phi(X_m^{i,\boldsymbol{\alpha}}, \frac{1}{N}\sum_{i=0}^N \delta_{X_m^{i,\alpha}}) A_m^{i,\boldsymbol{\alpha}} \alpha_m^i\right] \\ &= \mathbb{E}\left[\sum_{m=0}^T \frac{1}{N}\sum_{i=1}^N \Phi(X_m^{i,\boldsymbol{\alpha}}, \frac{1}{N}\sum_{i=0}^N \delta_{X_m^{i,\alpha}}) A_m^{i,\boldsymbol{\alpha}} \alpha_m^i\right] \\ &= \mathbb{E}\left[\sum_{m=0}^T \sum_{(x,a)\in\mathcal{S}} \nu_m^{N,p}(x, a)\Phi(x, \nu_{X,m}^{N,p}) a p_m(x)\right] \\ &= \mathbb{E}\left[\sum_{m=0}^T \Psi(\nu_m^{N,p}, p_m(\nu_m^{N,p}))\right] \end{aligned} \tag{15}$$

The asymptotic problem is written as:

$$\begin{cases} X_0^\alpha \sim \mu_0, \qquad A_0^\alpha = 1 \\ \alpha_n \sim \pi_n(\cdot | X_n^\alpha) = Be(p_n(X_n^\alpha)) \\ A_{n+1}^\alpha = A_n^\alpha \cdot (1 - \alpha_n) \\ X_{n+1}^\alpha = \begin{cases} F(n, X_n^\alpha, \mathcal{L}(X_n^\alpha), \epsilon_{n+1}), & \text{if } A_n^\alpha \cdot (1 - \alpha_n) = 1 \\ X_n^\alpha, & \text{otherwise,} \end{cases} \end{cases} \tag{16}$$

where the social cost is defined as:

$$J(p) := \sum_{m=n}^{T} \sum_{(x,a)\in\mathcal{S}} \nu_m^{p,\nu,n}(x,a) \Phi(x, \nu_{X,m}^p) a p_m(x)$$
$$= \sum_{m=n}^{T} \Psi(\nu_m^p, p_m(\nu_m^p)). \tag{17}$$

Let us recall that $\mathcal{P} := \{p : \{0, \ldots, T\} \times \mathcal{X} \times \mathcal{P}(\mathcal{S}) \to [0,1] : p \text{ is } L_p\text{-Lipschitz}\}$, the set of all possible admissible policies $p$. From now we are going to use the notation $\| \cdot \|$ for the norm associated to the total variation distance. Firstly we want to prove the at time time $n$ the distributions $\nu_m^{N,p}$ and $\nu_m^p$ are close in the following sense (see (Cui et al., 2023) for a similar setting).

**Lemma A.1** (Convergence of the measure). *Suppose Assumption 3.1 holds. Given the dynamics (14) and (16) for every $n = 0, \ldots, T$ it holds:*

$$\sup_{p\in\mathcal{P}} \mathbb{E}\left[\|\nu_n^{N,p} - \nu_n^p\|\right] = \mathcal{O}(1/\sqrt{N}). \tag{18}$$

*Proof.* We are going to follow an induction argument over the time steps:

*Initialization:* for $n = 0$, since we have indipendent samples at the starting point, by the law of large numbers (LLN) we have:

$$\sup_{p\in\mathcal{P}} \mathbb{E}\left[\|\nu_0^{N,p} - \nu_0^p\|\right] \to 0$$

with rate of convergence $\mathcal{O}\left(\frac{1}{\sqrt{N}}\right)$.

In particular, let us denote $\mathcal{S} := \{y_1, \cdots, y_K\}$, $\nu_0^p(y_i) = p_i$, $\nu_0^{N,p}(y_i) = \frac{1}{N}\sum_{i=1}^{N} \delta_{Y_i^\alpha}(y_i) = \frac{C(y_i)}{N}$, where $C(y_i)$ is defined as the number of agent that are in the state $y_i$ at time 0. We can write:

$$\mathbb{E}\left[\|\nu_0^{N,p} - \nu_0^p\|\right] = \frac{1}{2}\mathbb{E}\left[\sum_{i=1}^{|\mathcal{S}|} \left|\frac{C(y_i)}{N} - p_i\right|\right] \leq \frac{\sqrt{|\mathcal{S}|}}{2}\mathbb{E}\left[\sum_{i=1}^{|\mathcal{S}|}\left(\frac{C(y_i)}{N} - p_i\right)^2\right]^{1/2}$$

by Cauchy-Schwarz inequality. Notice now that $C(y_i) \sim \text{Bin}(N, p_i)$ and so

$$\sum_{i=1}^{|\mathcal{S}|} \text{Var}\left(\frac{C(y_i)}{N}\right) = \sum_{i=1}^{|\mathcal{S}|} \frac{p_i(1-p_i)}{N} = \frac{1 - \sum_{i=1}^{|\mathcal{S}|} p_i^2}{N} \leq \frac{|\mathcal{S}| - 1}{N|\mathcal{S}|}$$

since the quantity $1 - \sum_{i=1}^{|\mathcal{S}|} p_i^2$ has its max when $p_i = \frac{1}{|\mathcal{S}|}$.

Eventually, we obtain the explicit constant:

$$\mathbb{E}\left[\|\nu_0^{N,p} - \nu_0^p\|\right] \leq \frac{\sqrt{|\mathcal{S}| - 1}}{2\sqrt{N}}.$$

*Remark* A.2. Notice that the bound depends on the cardinality of the state space: more states lead to a larger upper bound, meaning possibly a larger discrepancy between the empirical and mean-field distributions. This is due to the fact that we used the total variation distance as metric, which sums over all possible states. In continuous space, this metric is not feasible and so usually the Wasserstein distance is used for convergence analysis (see (Carmona & Delarue, 2018)). Actually, in the finite space and discrete time setting, we have the following inequality:

$$d_{\min}\|\mu - \nu\|_{\text{TV}} \leq W_1(\mu, \nu) \leq D\|\mu - \nu\|_{\text{TV}},$$

where $d_{\min} := \min_{x \neq y} d(x, y)$ and $D := \max_{x \neq y} d(x, y)$. Notice that the Wasserstein distance in finite space and discrete time is defined as:

$$W_p(\mu, \nu) = \left( \min_{T \in \mathcal{C}(\mu, \nu)} \sum_{i=1}^{n} \sum_{j=1}^{n} d(x_i, x_j)^p \cdot T_{i,j} \right)^{\frac{1}{p}},$$

where $\mathcal{C}(\mu, \nu)$ is the set of couplings defined as:

$$\mathcal{C}(\mu, \nu) = \left\{ T \in \mathbb{R}^{n \times n} \;\middle|\; \sum_{j=1}^{n} T_{i,j} = \mu_i, \sum_{i=1}^{n} T_{i,j} = \nu_j, \; T_{i,j} \geq 0, \; \forall i, j \right\},$$

and $d(x_i, x_j)$ is the distance between points $x_i$ and $x_j$ in the metric space. More details on Wasserstein distances are described in (Villani, 2009) and (Arjovsky et al., 2017).

*Induction step:* assume now that (18) holds at time $n$. Using triangle inequality, at time $n + 1$ we have, for any $p \in \mathcal{P}$,

$$\mathbb{E}\left[\|\nu_{n+1}^{N,p} - \nu_{n+1}^p\|\right] \leq \mathbb{E}\left[\|\nu_{n+1}^{N,p} - \bar{F}(\nu_n^{N,p}, p_n(\nu_n^{N,p}))\|\right] + \mathbb{E}\left[\|\bar{F}(\nu_n^{N,p}, p_n(\nu_n^{N,p})) - \nu_{n+1}^p\|\right]$$

where we recall the expression of $\bar{F}$ described by (8).

For the first term, we have:

$$
\mathbb{E}\left[\left\|\nu_{n+1}^{N,p} - \bar{F}(\nu_n^{N,p}, p_n(\nu_n^{N,p}))\right\|\right]
$$
$$
= \mathbb{E}\left[\left\|\frac{1}{N}\sum_{i=1}^{N}\delta_{Y_{n+1}^{i,\alpha}} - \bar{F}(\nu_n^{N,p}, p_n(\nu_n^{N,p}))\right\|\right]
$$
$$
= \frac{1}{2}\mathbb{E}\left[\sum_{y \in \mathcal{S}}\left|\frac{1}{N}\sum_{i=1}^{N}\delta_{Y_{n+1}^{i,\alpha}}(y) - \bar{F}(\nu_n^{N,p}, p_n(\nu_n^{N,p}))(y)\right|\right]
$$
$$
= \frac{1}{2}\sum_{y \in \mathcal{S}}\mathbb{E}\left[\left|\frac{1}{N}\sum_{i=1}^{N}\delta_{Y_{n+1}^{i,\alpha}}(y) - \bar{F}(\nu_n^{N,p}, p_n(\nu_n^{N,p}))(y)\right|\right]
$$
$$
= \frac{1}{2}\sum_{y \in \mathcal{S}}\mathbb{E}\left[\mathbb{E}\left[\left|\frac{1}{N}\sum_{i=1}^{N}\delta_{Y_{n+1}^{i,\alpha}}(y) - \bar{F}(\nu_n^{N,p}, p_n(\nu_n^{N,p}))(y)\right| \;\middle|\; \mathbf{Y}_n^{\boldsymbol{\alpha}}\right]\right]
$$

The interpretation of $\bar{F}$ gives us:

$$
\bar{F}(\nu_n^{N,p}, p_n(\nu_n^{N,p}))(y) = \sum_{y'}\nu_n^{N,p}(y')\mathbb{P}(Y_{n+1}^p = y|Y_n^p = y')
$$
$$
= \frac{1}{N}\sum_{i=1}^{N}\mathbb{P}(Y_{n+1}^p = y|Y_n^p = Y_n^{i,p})
$$
$$
= \frac{1}{N}\sum_{i=1}^{N}\mathbb{P}(Y_{n+1}^{i,p} = y|Y_n^{i,p})
$$
$$
= \frac{1}{N}\sum_{i=1}^{N}\mathbb{E}[\delta_{Y_{n+1}^{i,p}}(y)|Y_n^{i,p}]
$$

where we used that the $i$ particles are indistinguishable and have the same transition functions.

So we can conclude the argument as:

$$\mathbb{E}\left[\left\|\nu_{n+1}^{N,p} - \bar{F}(\nu_n^{N,p}, p_n(\nu_n^{N,p}))\right\|\right] = \frac{1}{2}\sum_{y\in\mathcal{S}}\mathbb{E}\left[\mathbb{E}\left[\left\|\frac{1}{N}\sum_{i=1}^N \delta_{Y_{n+1}^{i,\alpha}}(y) - \mathbb{E}\left[\frac{1}{N}\sum_{i=1}^N \delta_{Y_{n+1}^{i,\alpha}}(y)\Big|\boldsymbol{Y}_n^{\boldsymbol{\alpha}}\right]\right\|\Big|\boldsymbol{Y}_n^{\boldsymbol{\alpha}}\right]\right] \le \frac{|S|}{4\sqrt{N}}$$

by the LLN, where again the bound is independent of $p \in \mathcal{P}$.

Indeed, given the past history $\boldsymbol{Y}_n^{\boldsymbol{\alpha}}$ the random variables $\delta_{Y_{n+1}^{i,\alpha}}$ become conditionally independent for every $i = 1,\ldots,N$. Furthermore each $\delta_{Y_{n+1}^{i,\alpha}}(y)$ is a Bernoulli random variable, therefore its variance $Var(\delta_{Y_{n+1}^{i,\alpha}}(y)|\boldsymbol{Y}_n^{\boldsymbol{\alpha}}) \le \frac{1}{4}$. Summing over all agents, the variance of the empirical mean becomes $\frac{1}{4N}$. Using Cauchy-Schwarz inequality, for any random variable $Z$ with finite variance $\mathbb{E}[|Z - \mathbb{E}[Z]|] \le \sqrt{Var(Z)}$, so in our case we obtained the constant $\frac{|S|}{4\sqrt{N}}$.

For the second term, by Lipschitz property of $\bar{F}$ and $p(\nu)$, we can write :

$$\begin{aligned}
\mathbb{E}&\left[\|\bar{F}(\nu_n^{N,p}, p_n(\nu_n^{N,p})) - \nu_{n+1}^p\|\right]\\
&= \mathbb{E}\left[\|\bar{F}(\nu_n^{N,p}, p_n(\nu_n^{N,p})) - \bar{F}(\nu_n^p, p_n(\nu_n^p))\|\right]\\
&\le L_{\bar{F}}\mathbb{E}\left[\|\|\nu_n^{N,p} - \nu_n^p\| + \|p_n(\nu_n^{N,p}) - p_n(\nu_n^p)\|\|\right]\\
&\le L_{\bar{F}}\mathbb{E}\left[\|\|\nu_n^{N,p} - \nu_n^p\| + L_p\|\nu_n^{N,p} - \nu_n^p\|\|\right]\\
&= (L_{\bar{F}}(1 + L_p))\mathbb{E}\left[\|\nu_n^{N,p} - \nu_n^p\|\right]\\
&\le \frac{|S|}{4\sqrt{N}}\left(\frac{1 - K^{n+1}}{1 - K}\right) + K^{n+1}\frac{\sqrt{|S| - 1}}{2\sqrt{N}} - \frac{|S|}{4\sqrt{N}}
\end{aligned}$$

by induction step, where $K := L_{\bar{F}}(1 + L_p)$ and the upper bound is independent of $p \in \mathcal{P}$ (since the constant $L_p$ is the same for all the control $p \in \mathcal{P}$).

We have thus proved by induction that:

$$E\left[\|\nu_{n+1}^{N,p} - \nu_{n+1}^p\|\right] \le \frac{|S|}{4\sqrt{N}}\left(\frac{1 - (L_{\bar{F}}(1 + L_p))^{n+1}}{1 - (L_{\bar{F}}(1 + L_p))}\right) + (L_{\bar{F}}(1 + L_p))^{n+1}\frac{\sqrt{|S| - 1}}{2\sqrt{N}},$$

for every time step $n = 0,\ldots,T$. $\qquad\square$

This result allows us to prove the following main theorem on the optimal cost approximation in the $N$-agent problem. This is a precise version of the informal statement in Theorem 3.2.

**Theorem A.3** ($\varepsilon$-approximation of the $N$-agent problem). *Suppose Assumption 3.1 holds. Given the dynamics (14) and (16) and the social cost associated (15), (17), let us denote by $p^*$ the optimal policy for the mean-field problem and by $\hat{p}$ the optimal policy for the $N$-agent problem. It holds:*

$$J^N(p^*,\ldots,p^*) - J^N(\hat{p},\ldots,\hat{p}) = \mathcal{O}(1/\sqrt{N}). \tag{19}$$

*Proof.* We can write:

$$J^N(p^*,\ldots,p^*) - J^N(\hat{p},\ldots,\hat{p}) = \left(J^N(p^*,\ldots,p^*) - J(p^*)\right) + \left(J(p^*) - J(\hat{p})\right) + \left(J(\hat{p}) - J^N(\hat{p})\right)$$

Notice first that we can bound this term simply deleting the second term in the r.h.s noticing $J(p^*) - J(\hat{p}) \le 0$ since $p^*$ is

optimal for the *mean-field* cost $J(p)$. For the first term we can write:

$$
\begin{aligned}
J^N(p^*, & \ldots, p^*) - J(p^*) \\
&= \mathbb{E}\left[\sum_{m=0}^T \Psi(\nu_m^{N,p^*}, p_m^*(\nu_m^{N,p^*}))\right] - \sum_{m=n}^T \Psi(\nu_m^{p^*}, p_m^*(\nu_m^{p^*})) \\
&= \sum_{n=0}^T \mathbb{E}\left[\Psi(\nu_n^{N,p^*}, p_n^*(\nu_n^{N,p^*})) - \Psi(\nu_n^{p^*}, p_n^*(\nu_n^{p^*}))\right] \\
&\leq L_\Psi \sum_{n=0}^T \mathbb{E}\left[\left\|\nu_n^{N,p^*} - \nu_n^{p^*}\right\| + \left\|p_n^*(\nu^{N,p^*}) - p_n^*(\nu_n^{p^*})\right\|\right] \\
&\leq L_\Psi(1 + L_p) \sum_{n=0}^T \mathbb{E}\left[\left\|\nu_n^{N,p^*} - \nu_n^{p^*}\right\|\right] \\
&\leq TL_\Psi(1 + L_p) \sup_{n \in \{0,\ldots,T\}} \mathbb{E}\left[\left\|\nu_n^{N,p^*} - \nu_n^{p^*}\right\|\right] \\
&\leq TL_\Psi(1 + L_p)\left[\frac{|S|}{4\sqrt{N}}\left(\frac{1 - (L_{\bar{F}}(1 + L_p))^T}{1 - (L_{\bar{F}}(1 + L_p))}\right) + (L_{\bar{F}}(1 + L_p)))^T \frac{\sqrt{|S| - 1}}{2\sqrt{N}}\right],
\end{aligned}
$$

by Lemma A.1. For the last term $J(\hat{p}) - J^N(\hat{p})$, we can apply the same argument that we just described. Similarly, we obtain:

$$
J^N(p^*, \ldots, p^*) - J^N(\hat{p}, \ldots, \hat{p}) \leq 2TL_\Psi(1 + L_p)\left[\frac{|S|}{4\sqrt{N}}\left(\frac{1 - (L_{\bar{F}}(1 + L_p))^T}{1 - (L_{\bar{F}}(1 + L_p))}\right) + (L_{\bar{F}}(1 + L_p)))^T \frac{\sqrt{|S| - 1}}{2\sqrt{N}}\right]
$$

$\square$

## B. Proof of Theorem 4.1

Let us prove Theorem 4.1.

*Proof.* To prove this result, we will show that we can reduce the problem to a mean-field optimal control problem in discrete time and continuous space. Then we can apply the well-studied dynamic programming principle for mean-field Markov decision processes (MFMDPs) (see e.g. (Motte & Pham, 2022; Carmona et al., 2023; Bäuerle, 2023)). We have:

$$
\begin{aligned}
V_n(\nu) &= \inf_{p \in \mathcal{P}_{n,T}} \sum_{m=n}^T \sum_{(x,a) \in \mathcal{S}} \nu_m^{p,\nu,n}(x,a)\Phi(x, \mu_m^{p,\nu,n})ap_m(x) \\
&= \inf_{p \in \mathcal{P}_{n,T}} \sum_{m=n}^T \Psi(\nu_m^{p,\nu,n}, p_m),
\end{aligned}
$$

where $\Psi : \mathcal{P}(\mathcal{X} \times \{0,1\}) \times \mathcal{P}(\mathcal{X}) \to \mathbb{R}$ and it is defined as:

$$
\Psi(\nu, q) := \sum_{(x,a) \in \mathcal{S}} \nu(x,a)\Phi(x, \nu_X)aq(x).
$$

Then we can define the process $Z$ taking value in $\mathcal{P}(\mathcal{X} \times \{0,1\})$:

$$
Z_n^p = z = \nu; \qquad Z_m^p := \nu_m^{p,\nu,n} \quad \forall m \geq n
$$

such that it follows the dynamics $Z_{m+1}^p = \bar{F}(Z_m^p, p_m)$ for every $m = n, \ldots, T - 1$. We can write:

$$
V_n(z) = \inf_{p \in \mathcal{P}_{n,T}} \sum_{m=n}^T \Psi(Z_m^p, p_m),
$$

and we recognize a well-studied control problem for which the DPP is:

$$V_n(z) = \inf_{h \in \mathcal{H}} \Psi(z, h) + V_{n+1}(\bar{F}(z, h)).$$

where $\mathcal{H}$ is the set of all functions $h : \mathcal{X} \to [0, 1]$. Finally, we can recover our result:

$$V_n(\nu) = \inf_{h \in \mathcal{H}} \sum_{(x,a) \in \mathcal{S}} \nu(x, a)\Phi(x, \nu_X)ah(x) + V_{n+1}(\bar{F}(\nu, h)). \tag{20}$$

where $\nu_X$ is the first marginal of the distribution $\nu$. □

## C. Algorithms

Alg. 3 and 4 present respectively the direct approach and the DP-based method.

---

**Algorithm 3** Direct Approach for MFOS

---

**Require:** Time-dependent stopping decision neural network: $\psi_\theta : \{0, \dots, T\} \times \mathcal{X} \times \mathcal{P}(\mathcal{S}) \to [0, 1]$, cost function $\Phi$, mean-field dynamic transition $\bar{F}$, time horizon $T$, max training iteration $N_{\texttt{iter}}$.
    **// TRAINING**
1: **for** $k = 0, \dots, N_{\texttt{iter}} - 1$ **do**
2:     Uniformly sample initial distribution $\nu_0^p$ from the probability simplex on $\mathbb{R}^{2|\mathcal{X}|}$
3:     **for** $n = 0, \dots, T$ **do**
4:         $p_n(x) = \psi_\theta(x, \nu_n^p, n; \theta_k)$ for any $x \in \mathcal{X}$     ▷ Compute stopping probability
5:         $\ell_n = \sum_{x \in \mathcal{X}} \nu_n^p(x, 1)\Phi(x, \mu_n)p_n(x)$     ▷ Compute loss at time $n$
6:         $\nu_{n+1}^p = \bar{F}(\nu_n^p, p_n)$     ▷ Simulate MF dynamic
7:     $\ell = \sum_{n=0}^{T} \ell_n$     ▷ Compute the total loss
8:     $\theta_{k+1} = \texttt{optimizer\_update}(\theta_k, \ell(\theta_k))$     ▷ AdamW optimizer step
9: Set $\theta^* = \theta_{N_{\texttt{iter}}}$
10: **return** $\psi_{\theta^*}$

---

**Algorithm 4** Dynamic Programming Approach for MFOS

---

**Require:** A sequence of stopping decision neural network: $\psi_\theta^n : \mathcal{X} \times \mathcal{P}(\mathcal{S}) \to [0, 1]$ for $n \in \{0, \dots, T - 1\}$, cost function $\Phi$, mean-field dynamic transition $\bar{F}$, time horizon $T$, max training iteration $N_{\texttt{iter}}$.
    **// TRAINING**
1: Set $\psi_\theta^T = 1$ since all distribution stopped at time $T$.
2: **for** $n = T - 1, \dots, 0$ **do**     ▷ Train backward in time
3:     **for** $k = 0, \dots, N_{\texttt{iter}} - 1$ **do**
4:         Uniformly sample initial distribution $\nu_n^p$ from the probability simplex on $\mathbb{R}^{2|\mathcal{X}|}$
5:         **for** $m = n, \dots, T$ **do**
6:             **if** $m = n$ **then**
7:                 $p_m(x) = \psi_\theta^m(x, \nu_m^p; \theta_k^n)$     ▷ Compute with NN for current time
8:             **else**
9:                 $p_m(x) = \psi_\theta^m(x, \nu_m^p; \theta^{m,*})$     ▷ Compute with trained NN from future time
10:             $\ell_m = \sum_{x \in \mathcal{X}} \nu_m^p(x, 1)\Phi(x, \mu_m)p_m(x)$     ▷ Compute loss at time $m$
11:             $\nu_{m+1}^p = \bar{F}(\nu_m^p, p_m)$     ▷ Simulate MF dynamic
12:         $\ell = \sum_{m=n}^{T} \ell_m$     ▷ Compute the total loss from time $n$ to $T$
13:         $\theta_{k+1}^n = \texttt{optimizer\_update}(\theta_k^m, \ell(\theta_k^n))$     ▷ AdamW optimizer step
14:     Set $\theta^{n,*} = \theta_{N_{\texttt{iter}}}^n$     ▷ Stored trained weight

---

## D. Implementation details

In this section, we will discuss the choice of neural networks, training batch size, learning rate, and iterations, and all the related hyperparameters as well as computational resources used.

**Neural Network Architectures:** We have 4 variants of neural networks.

For the direct approach, the neural network takes an input time $t$, while for the DPP approach, the neural network does not need time input.

For the asynchronous stopping problem, besides time, the neural network has two spatial inputs 1) the state $x$, represented as an integer, goes through an embedding layer with learnable parameters and the results are fed to other operations. 2) the distribution $\nu$, represented as a vector, is inputted to the neural net directly. For the synchronous stopping problem, the neural network only has one spatial input, which is the distribution $\nu$, and is treated as the same way as discussed before.

In general, our neural network has the following structure. Our neural network takes an input pair $(x, t)$, where $x$ is the spatial input, $t$ is the time. If $t$ is a needed input, then it is passed through a module to generate a standard sinusoidal embedding and then fed to 2 fully connected layers with Sigmoid Linear Unit (SiLU) and generate an output $t_{\text{out}}$. Spatial input $x$ is passed through an MLP with $k$ residual blocks, each containing 4 linear layers with hidden dimension $D$ and SiLU activation. This generates an output $y_{\text{out}}$. Our final output out is computed through,

$$\text{out} = \text{Outmod}(\text{GroupNorm}(y_{\text{out}} + t_{\text{out}}))$$

where Outmod is an out module that consists of 3 fully connected layers with hidden dimension $D$ and SiLU activation, GroupNorm stands for group normalization. If $t$ is not a needed input, then set $t_{\text{out}} = 0$.

For all the test cases we have experimented with, we use $k = 3$, $D = 128$ for all the 1D experiments and $k = 5$, $D = 256$ for the 2D experiments.

**Computational Resources:** We run all the numerical experiments on an RTX 4090 GPU and a MacBook Pro with M2 Chip. For any of the test cases, one run took at most 3 minutes on GPUs and 7 minutes on CPUs.

**Training Hyperparameters:** For all the experiments, we choose an initial learning rate $10^{-4}$ of the AdamW optimizer. Each training is at most $10^4$ iterations, with a batch size 128. The number of training iterations is chosen based on numerical evidence and trial and error. We start with a moderate number and then increase it if the model shows signs of undertraining and is far from convergence.

## E. Numerical Experiments details

This section aims to complete the results of the 6 numerical experiments conducted. While some of the following plots have been previously discussed in Section 6, we provide the full descriptions of Example E.1 and Example E.2 here for the sake of completeness.

### E.1. Example 1: Towards the Uniform

We take state space $\mathcal{X} = \{0, 1, 2, 3, 4\}$, time horizon $T = 4$, transition function $F(n, x, \mu, \epsilon) = x + 1$ which means that the agent deterministically moves to the state on the right, with boundary at $x = 4$ (meaning that once at 4, the agent does not move anymore), and cost function $\Phi(x, \mu) = \mu(x)$ which depends on the mean field only through the state of the agent (this is sometimes called local dependence). For the testing distribution, we take a distribution concentrated on state $x = 0$, denoted as $\mu_0 = \delta_0$. It can be seen that the optimal strategy consists in spreading the mass to make it as close as uniform as possible (hence the name of this example). Fig. 9 shows that the testing loss decays towards the true optimal value, and the distribution evolves towards a uniform distribution as expected. Fig. 10 shows the losses with DPP: there is one curve per time step. At time 0, the value is close to the optimal value. First, we explain how the optimal value is computed. Since the agents move deterministically to the right, the only option to freeze some mass at a state $x$ is to do it at time $n$. It can be seen that: for every $n = 0, \ldots, T$ and for every $x \in \mathcal{X}$, we want to have $p_n(x = n) = \frac{1}{T+1-n} \mathbb{1}_{x=n}$ for $n < T$ and $p_n(x) = 1$ for $n = T$. Actually notice that for all $x \neq n$ the choice of $p_n$ is arbitrary so, at every time-step $n$ we can apply the same $p_n$ for every state $x$. This brings us to optimize over the set of synchronous stopping times.

Then we can compute the optimal value and obtain: $V^{*, \delta_0} := \frac{T+2}{2(T+1)}$.

Figs. 11 and 12 show the result for synchronous stopping.

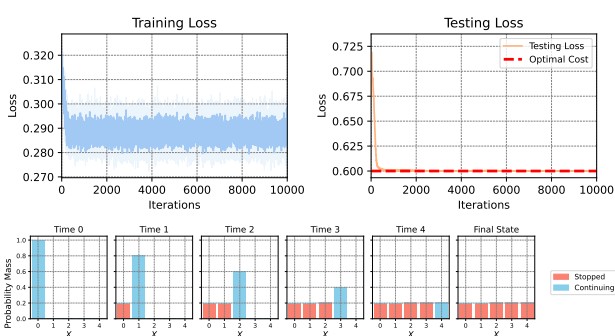

Figure 9: Example 1. DA results, asynchronous stopping. Top: training and testing losses. Bottom: evolution of the distribution after training.

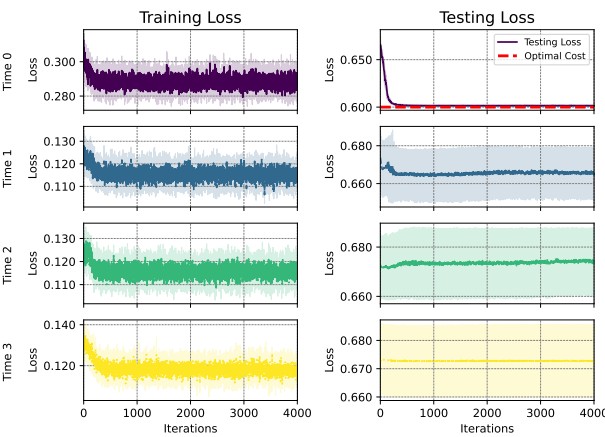

Figure 10: Example 1. DPP results, asynchronous stopping. Training and testing losses.

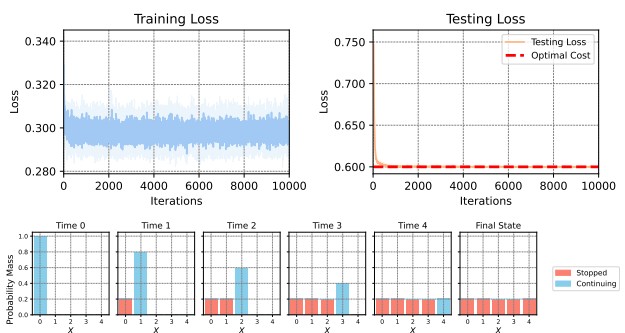

Figure 11: Example 1. DA results, synchronous stopping. Top: training and testing losses. Bottom: evolution of the distribution after training.

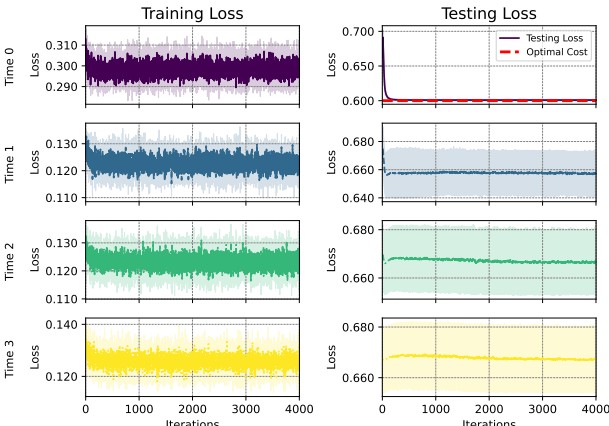

Figure 12: Example 1. DPP results, synchronous stopping. Training and testing losses.

### E.2. Example 2: Rolling a Die

In this example, at every time step, a fair six-sided die is rolled. This takes the role of the noise $\epsilon \sim \mathcal{U}(\mathcal{X})$ where $\mathcal{X} = \{1, 2, 3, 4, 5, 6\}$. The system starts in the initial distribution $\eta = \frac{1}{4}\delta_1 + \frac{1}{4}\delta_2 + \frac{1}{2}\delta_5$, and evolves according to the dynamics (5) with: $\mu_0 = \eta$, $F(n, x, \mu, \epsilon_{n+1}) = \epsilon_{n+1}$. The social cost function associated to this scenario is $\Phi(x, \mu) = x$. DA and DPP results are shown in Figs. 13 and 14 respectively. Again, we observe convergence to the true optimal value here. The optimal value is computed as follows. Using the dynamic programming principle described in (11), we can compute the optimal strategy and the optimal value:

$$p_0(\cdot) = (1, 0, 0, 0, 0, 0) \qquad p_1(\cdot) = (1, 1, 0, 0, 0, 0)$$
$$p_2(\cdot) = (1, 1, 0, 0, 0, 0) \qquad p_3(\cdot) = (1, 1, 0, 0, 0, 0)$$
$$p_4(\cdot) = (1, 1, 1, 0, 0, 0) \qquad p_5(\cdot) = (1, 1, 1, 1, 1, 1)$$

$$V^{*,\eta} = 1,6525.$$

For our considered initial distribution, this is one of the possible optimal strategies, since we have no mass on some states and thus can assign any stopping probability to them. However, the solution we have presented is the only optimal solution for all possible initial distributions. Note that if we optimize on the class of synchronous stop times, we do not reach

the same optimal value, but we reach a higher value, concluding that for this type of problem, it is better to optimize on asynchronous stop times. In fact, when you narrow the decision only to the class of synchronous stop times is better to stop everyone at the first initial state reaching a value of $\tilde{V}* = 3,25 > 1,6525 = V^*$. Synchronous stopping results are shown in Fig. 15 and 16.

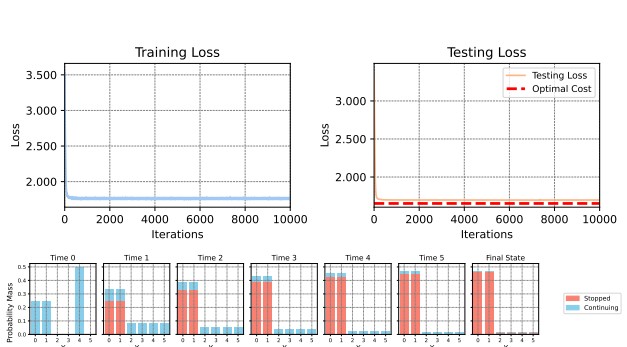

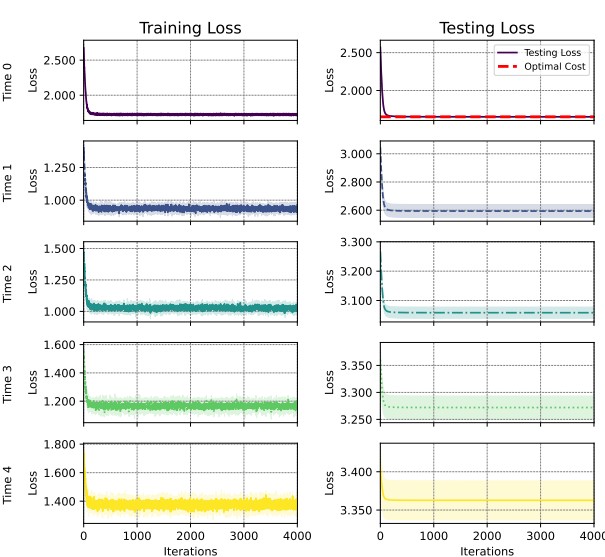

Figure 13: Example 2. DA results, asynchronous stopping. Top: training and testing losses. Bottom: evolution of the distribution after training.

Figure 14: Example 2. DPP results, asynchronous stopping. Training and testing losses.

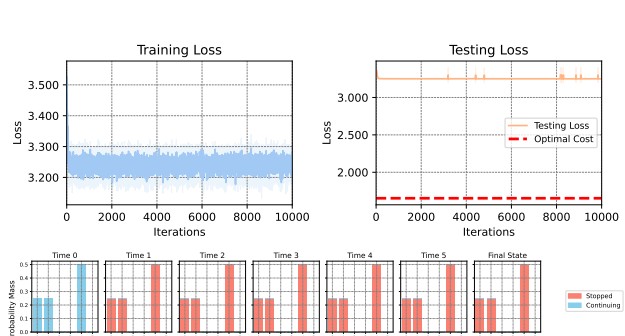

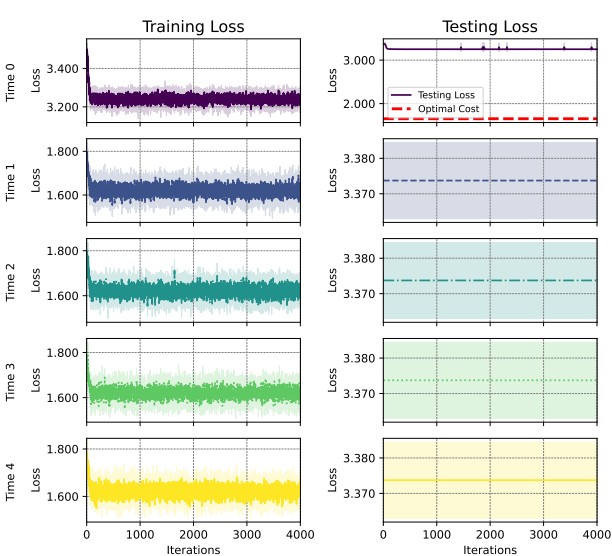

Figure 15: Example 2. DA results, synchronous stopping. Top: training and testing losses. Bottom: evolution of the distribution after training.

Figure 16: Example 2. DPP results, synchronous stopping. Training and testing losses.

### E.3. Example 3: Crowd Motion with Congestion.

This example extends the previous one, adding a congestion factor. The transition probabilities are:

$$p_n(z,x) := P(X_{n+1} = z | X_n = x) = \begin{cases} \frac{1}{6}(1 - \frac{1}{5}C_{\text{cong}}\mu(x)), & \text{if } z \neq x, \\ \frac{1}{6}(1 + C_{\text{cong}}\mu(x)), & \text{if } z = x. \end{cases} \quad (21)$$

Let us set $C_{\text{cong}} = 0.8$. However, the reasoning regarding the differences between scenarios in which the central planner optimizes the set of asynchronous stopping times or the set of synchronous stopping times is similar. DA testing and training losses are shown in Fig. 17. DPP testing and training losses are shown in Fig. 18.

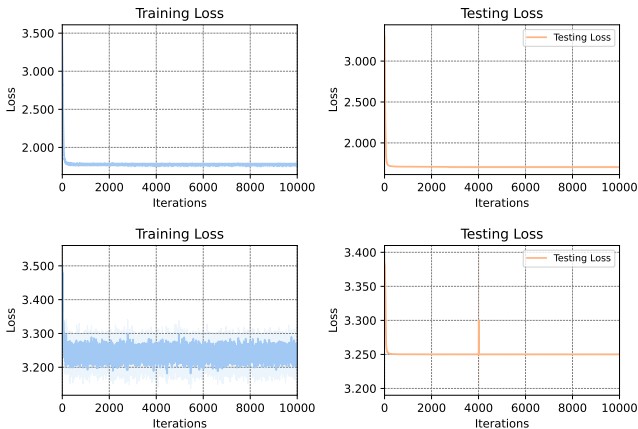

Figure 17: Example 3. DA results. Training and testing losses. Top: asynchronous stopping. Bottom: synchronous stopping.

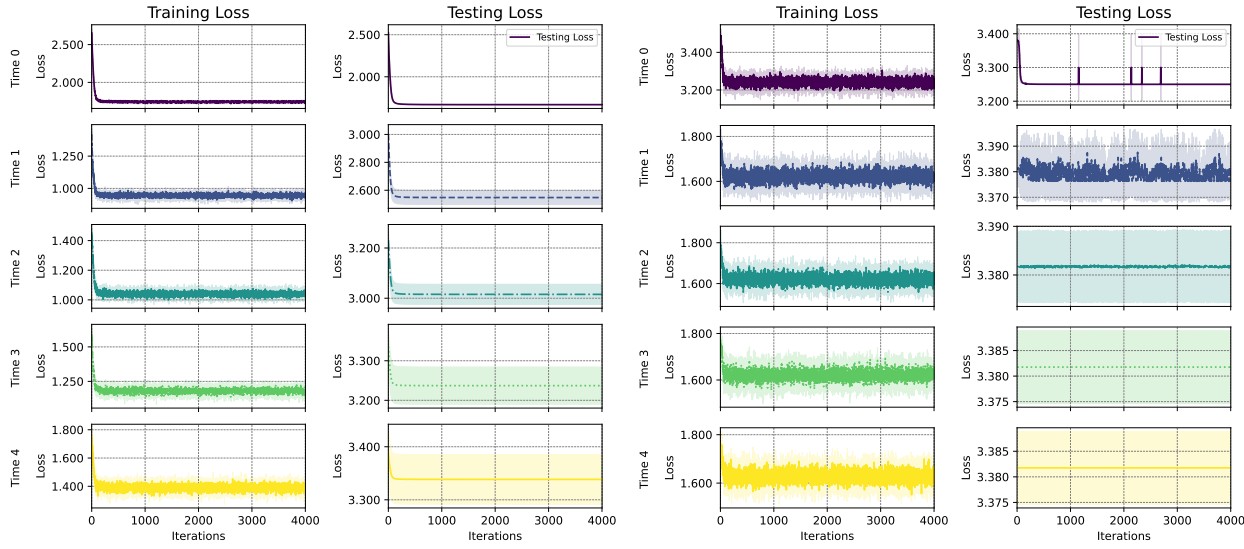

Figure 18: Example 3. DPP results. Training and testing losses. Left: asynchronous stopping. Right: synchronous stopping.

### E.4. Example 4: Distributional Cost

DA results are shown in Fig. 19 and 20. DPP results are shown in Fig. 21.

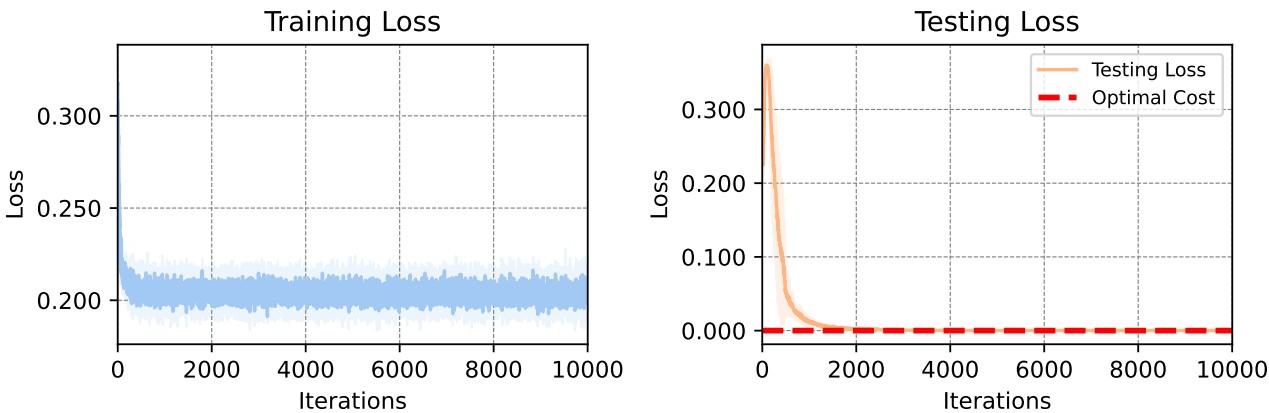

Figure 19: Example 4. DA results, asynchronous stopping. Training and testing losses.

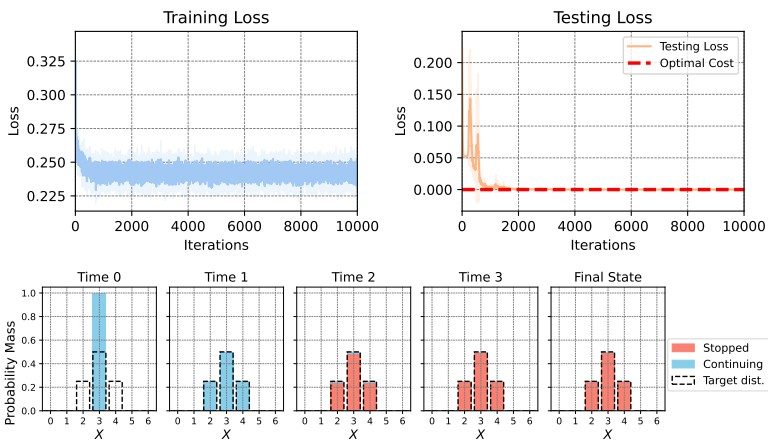

Figure 20: Example 4. DA results, synchronous stopping. Top: training and testing losses. Bottom: evolution of the distribution after training.

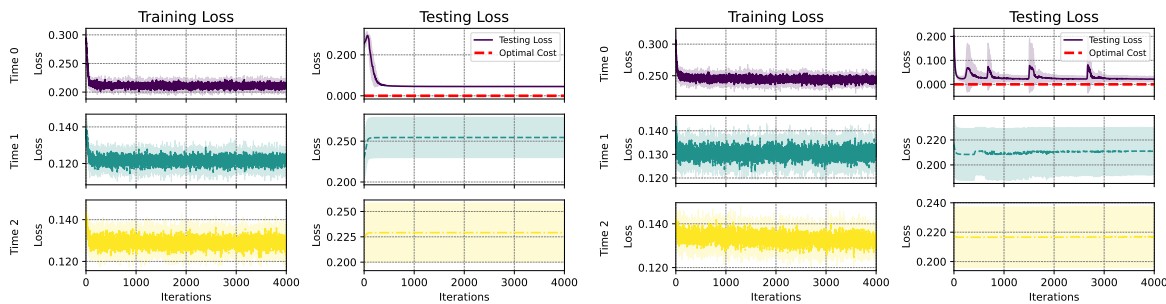

Figure 21: Example 4. DPP results, asynchronous and synchronous stopping. Training and testing losses.

### E.5. Example 5: Towards the Uniform in 2D

Asynchronous stopping results, including training losses, testing losses, distribution evolution, and stopping probability are shown in Figs. 22 and 23. Synchronous stopping results are shown in Figs. 24 and 25.

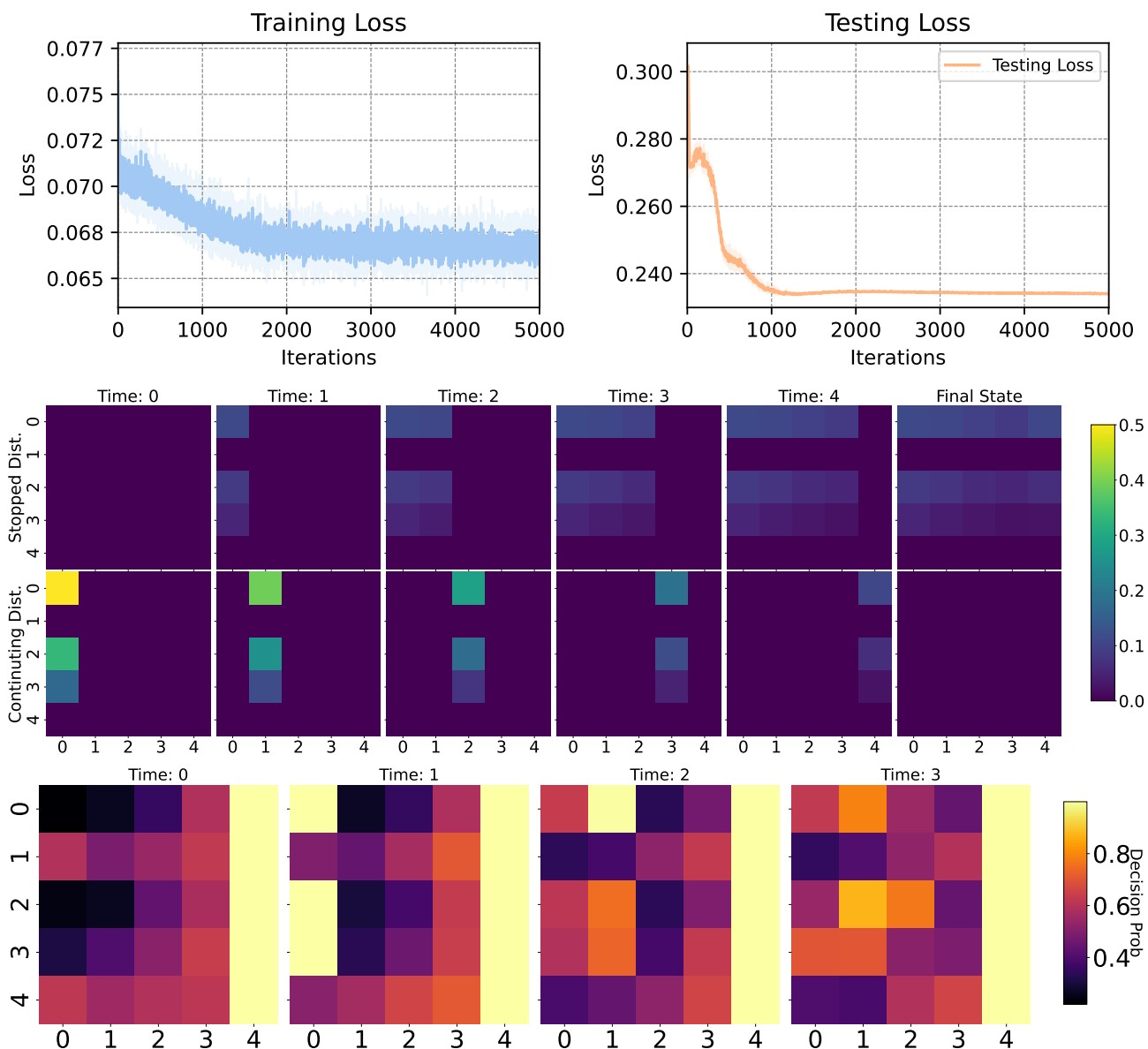

Figure 22: Example 5. DA results, asynchronous stopping. Top: training and testing losses. Bottom: evolution of the distribution and stopping probability after training.

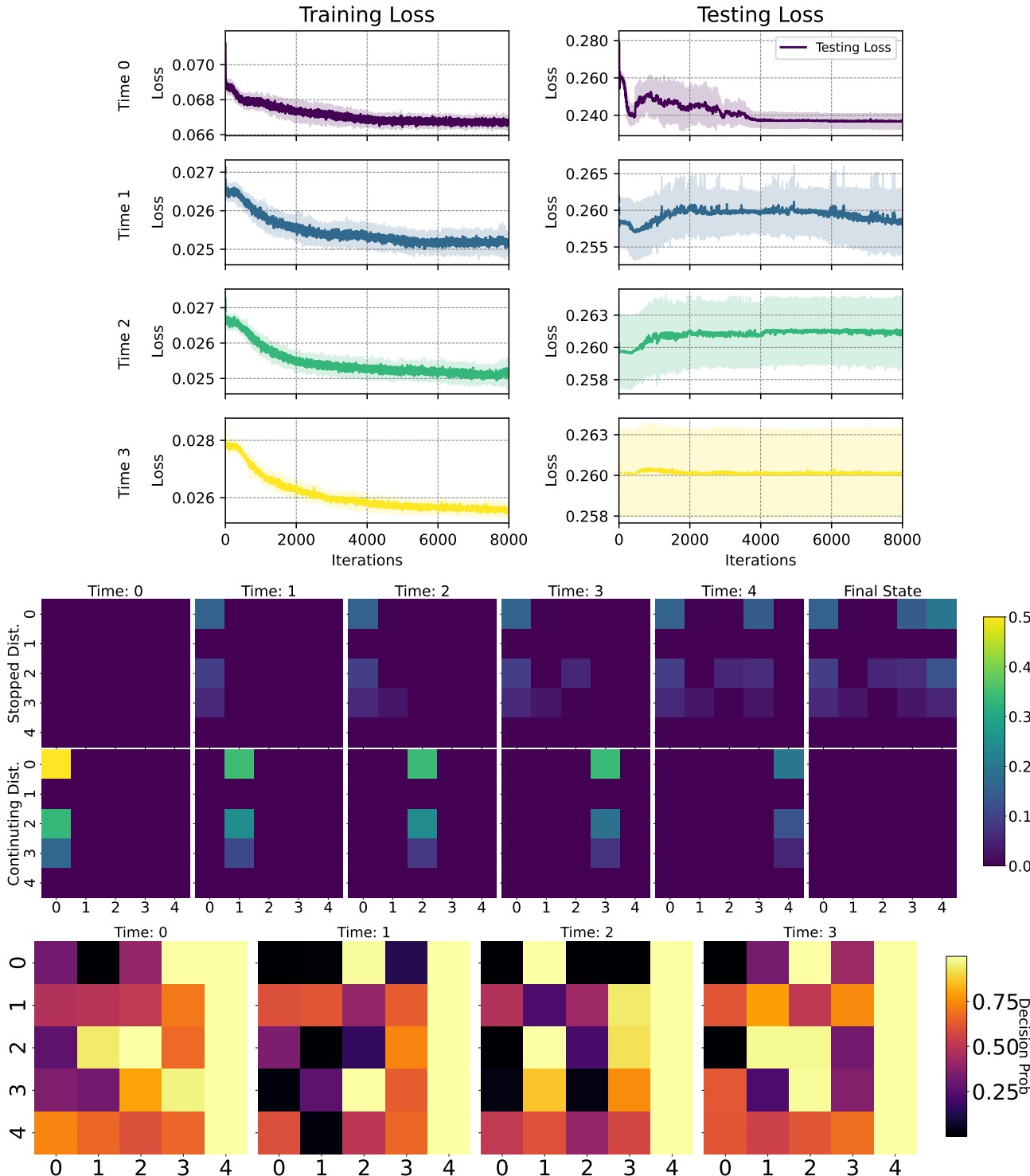

Figure 23: Example 5. DPP results, asynchronous stopping. Top: training and testing losses. Bottom: evolution of the distribution and stopping probability after training.

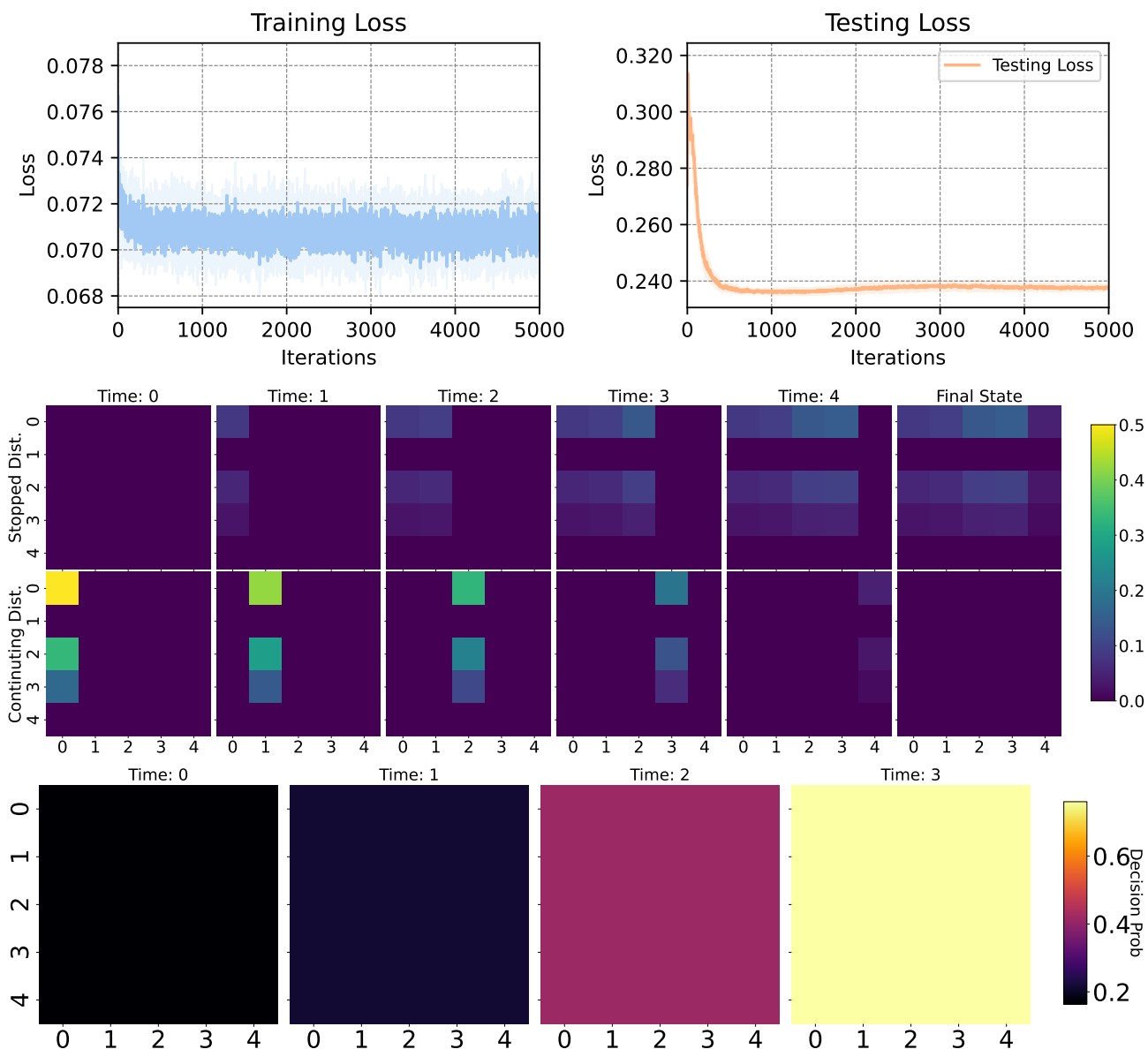

Figure 24: Example 5. DA results, synchronous stopping. Top: training and testing losses. Bottom: evolution of the distribution and stopping probability after training.

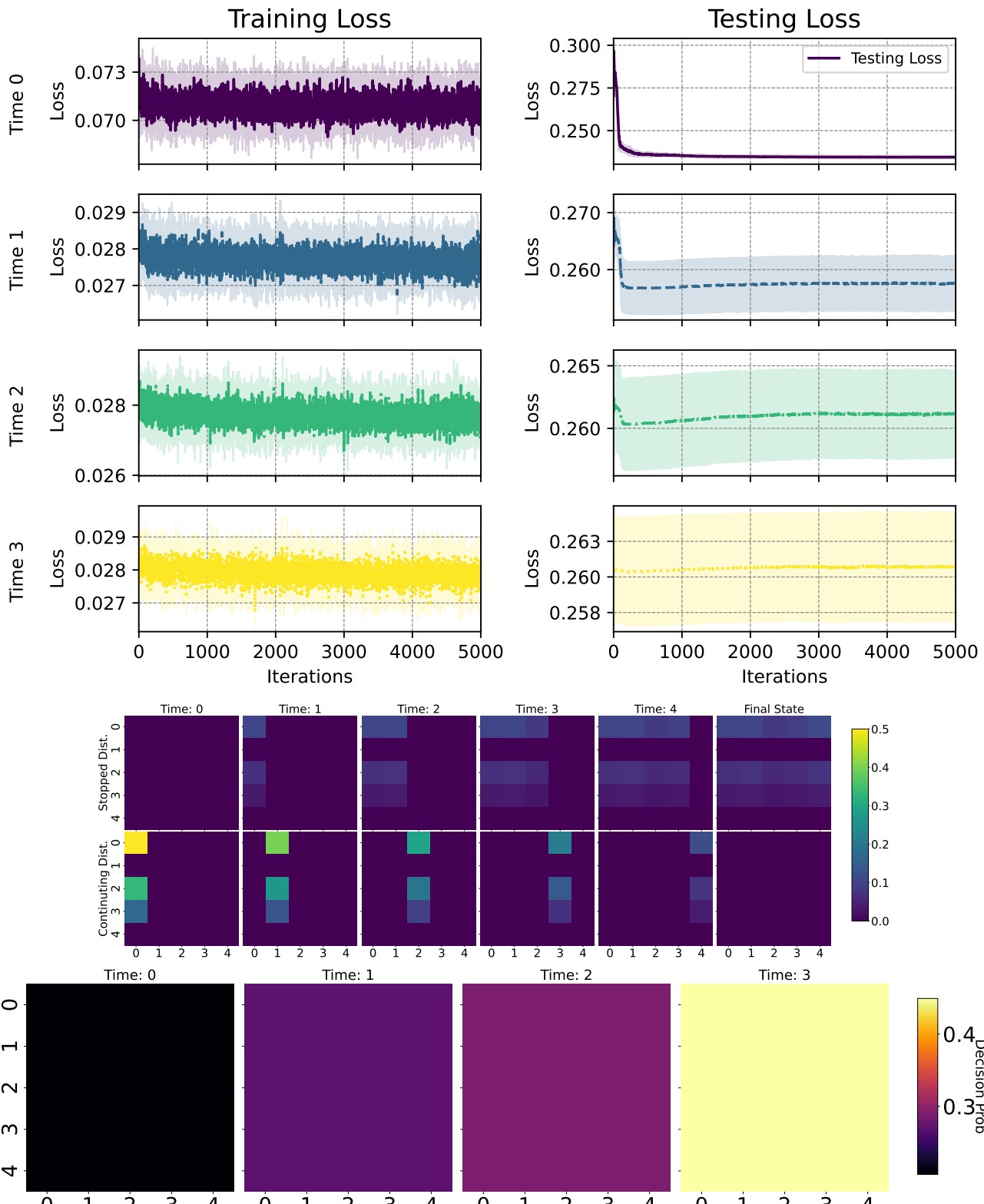

Figure 25: Example 5. DPP results, synchronous stopping. Top: training and testing losses. Bottom: evolution of the distribution and stopping probability after training.

### E.6. Example 6: Matching a Target with a Fleet of Drones.

In this example, we extend our framework by incorporating a terminal cost and common noise. This allows us to consider a richer and more realistic class of MFOS environments. We extend the dynamics defined in (5) in the following way:

$$
\begin{cases}
X_0^\alpha \sim \mu_0, \qquad A_0^\alpha = 1 \\
\alpha_n \sim \pi(\cdot | X_n^\alpha) = Be(p_n(X_n^\alpha)) \\
A_{n+1}^\alpha = A_n^\alpha \cdot (1 - \alpha_n) \\
X_{n+1}^\alpha = \begin{cases} F(n, X_n^\alpha, \mu_n^\alpha, \epsilon_{n+1}, \epsilon_{n+1}^0), & \text{if } A_n^\alpha \cdot (1 - \alpha_n) = 1 \\ X_n^\alpha, & \text{otherwise.} \end{cases}
\end{cases}
\tag{22}
$$

where $\epsilon_n^0$ is the common noise that affects the dynamics of all agents equally. Note that with the presence of a common noise, the mean-field distribution $\nu$ is not deterministic, but instead it is a random variable that evolves conditionally with respect to the common noise.

Furthermore the social cost defined in (9) can be extended by adding a terminal cost:

$$
J(p) = \mathbb{E}^0 \left[ \sum_{n=0}^{T} \sum_{(x,a) \in \mathcal{S}} \left( \nu_n^p(x,a) \Phi(x, \nu_{X,n}^p) a p_n(x) \right) + g(\nu_{X,T}^p) \right],
\tag{23}
$$

where $g : \mathcal{P}(\mathcal{X}) \to \mathbb{R}$ is the terminal cost and $\mathbb{E}^0$ is the expectation with respect the common noise realization.

The results for DA for different target distributions are provided in Fig. 26. The results for DPP for different target distributions are provided in Fig. 27.

It is evident that, unlike the DPP, the optimal strategy in the DA tends to stop with high probability at the final time steps, as clearly illustrated for the target distributions corresponding to the letters "O" and "S".

## F. Hyperparameters sweep

In this section, we show the results of a sweep over the learning rate for Example 1 with the two methods and the two types of stopping times. We consider learning rates $10^{-2}$, $10^{-3}$, and $10^{-4}$ in this order in the plots from top to bottom.

Direct method stopping: Figs. 28 and 29 show the losses for the asynchronous and the synchronous stopping times respectively.

Direct method stopping: Figs. 30 and 31 show the losses for the asynchronous and the synchronous stopping times respectively.

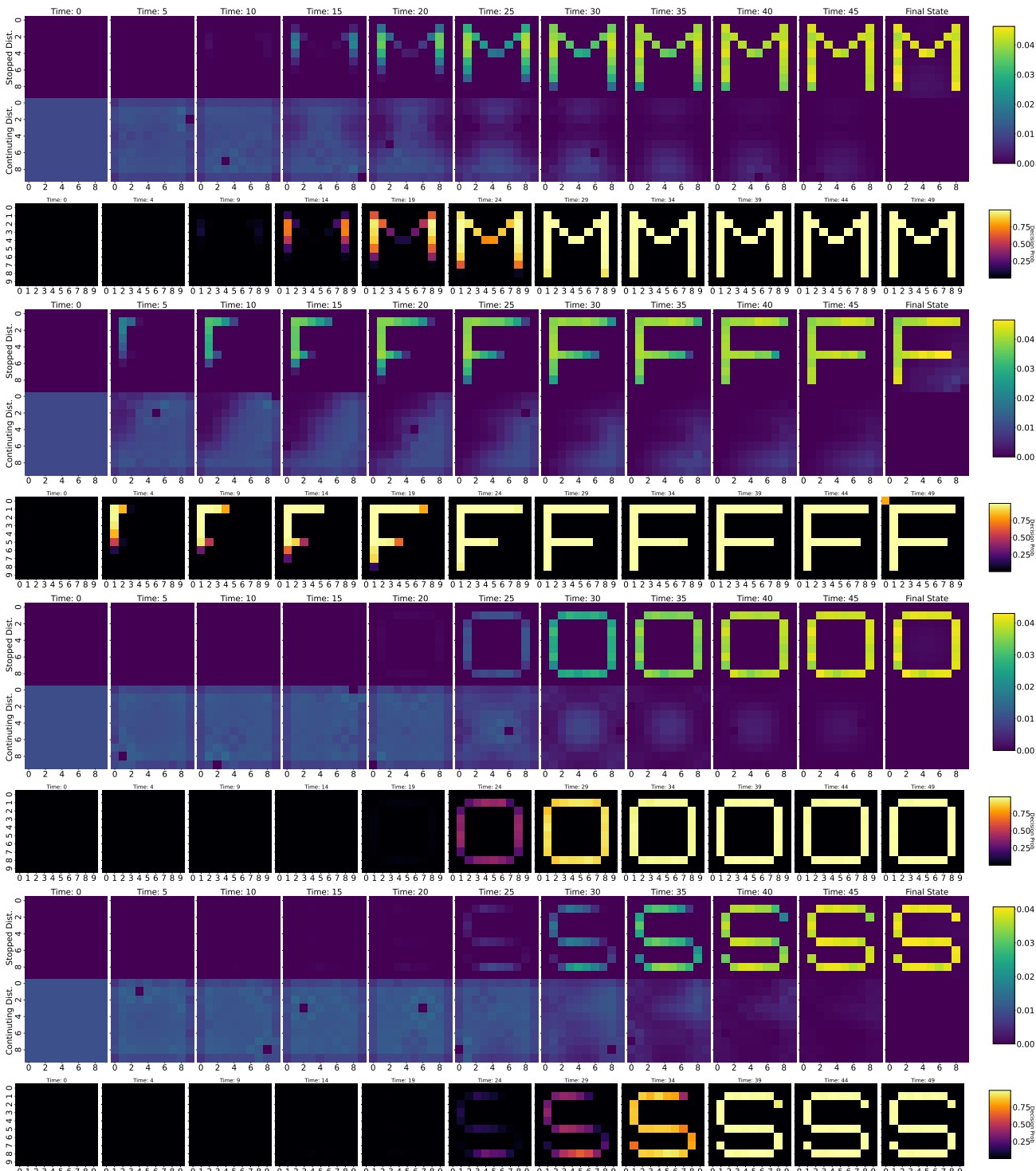

Figure 26: Example 6. DA results, asynchronous stopping. Match the Letter "M", "F", "O", "S", in a $10 \times 10$ grid with common noise.

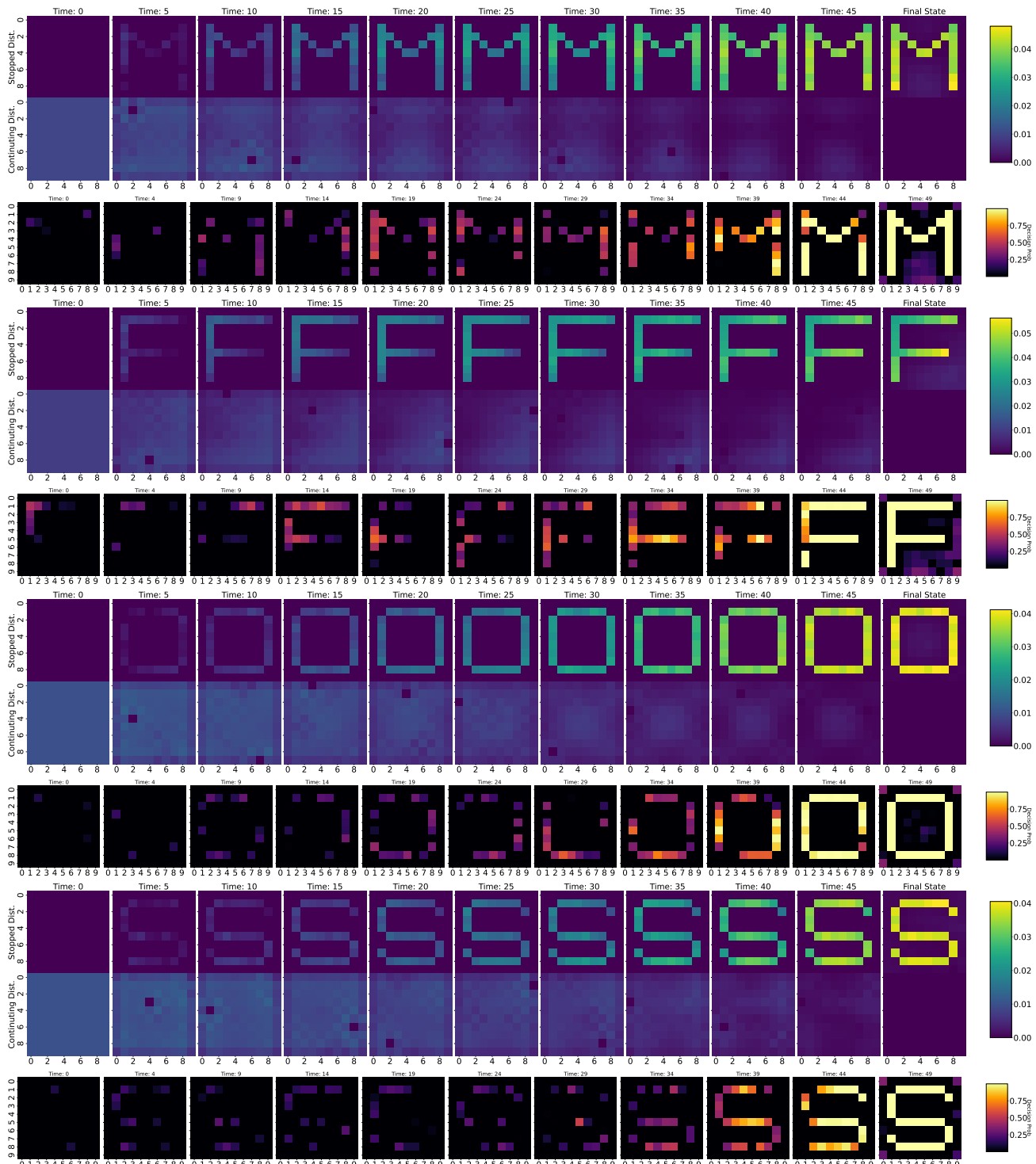

Figure 27: Example 6. DPP results, asynchronous stopping. Match the Letter "M", "F", "O", "S", in a $10 \times 10$ grid with common noise

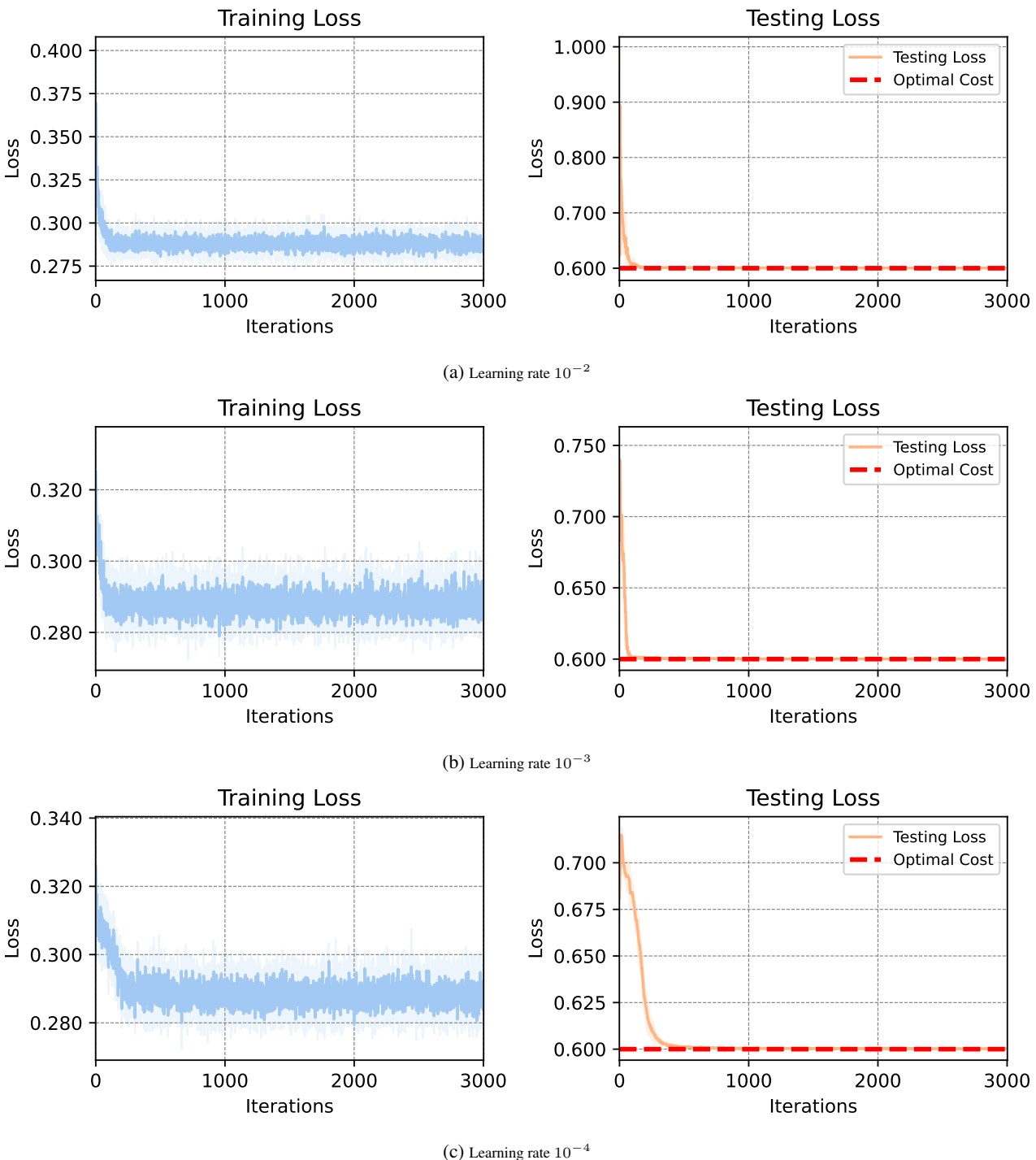

(a) Learning rate $10^{-2}$

(b) Learning rate $10^{-3}$

(c) Learning rate $10^{-4}$

Figure 28: Example 1: Sweep of learning rates. DA results, asynchronous stopping.

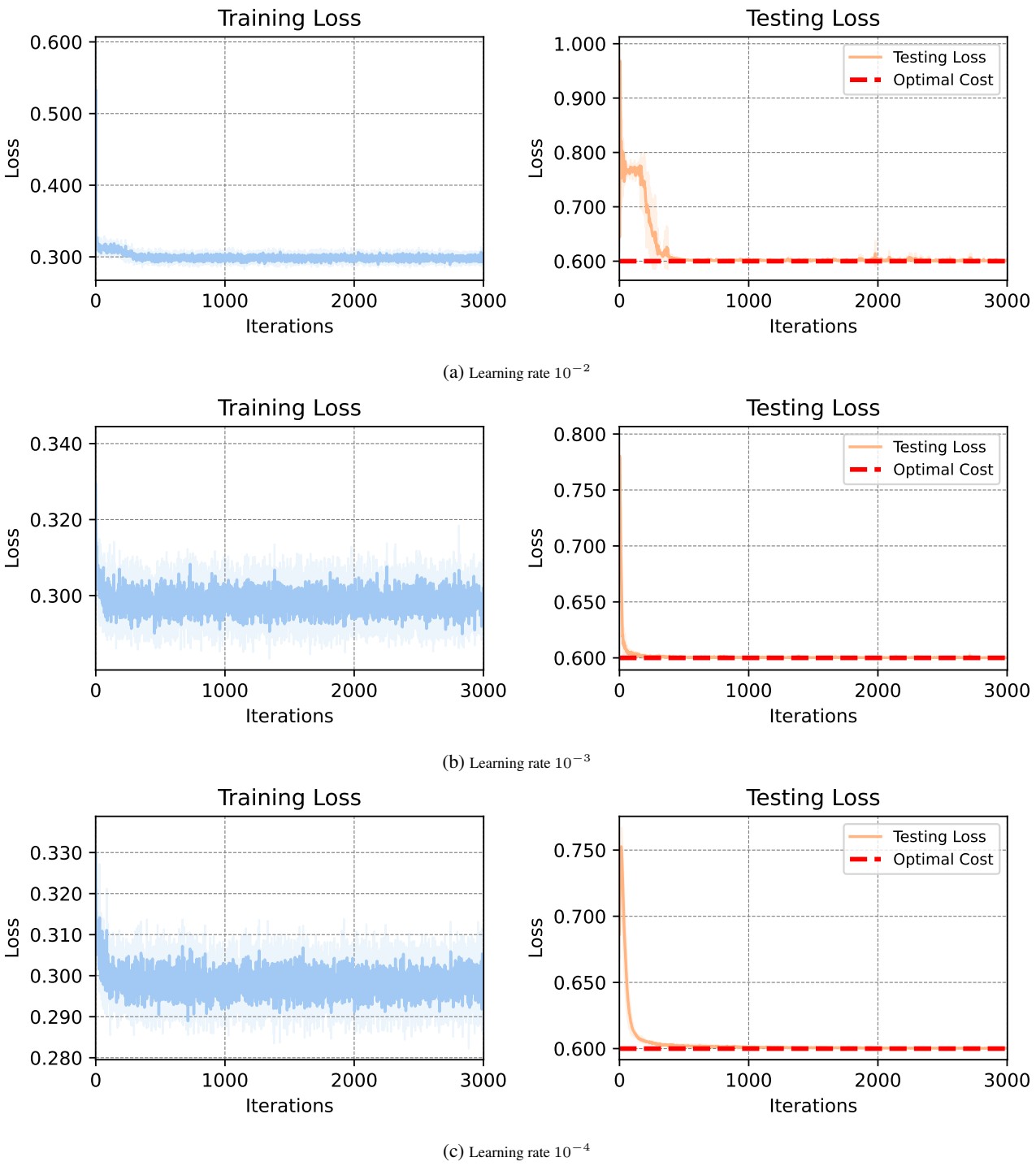

(a) Learning rate $10^{-2}$

(b) Learning rate $10^{-3}$

(c) Learning rate $10^{-4}$

Figure 29: Example 1: Sweep of learning rates. DA results, synchronous stopping.

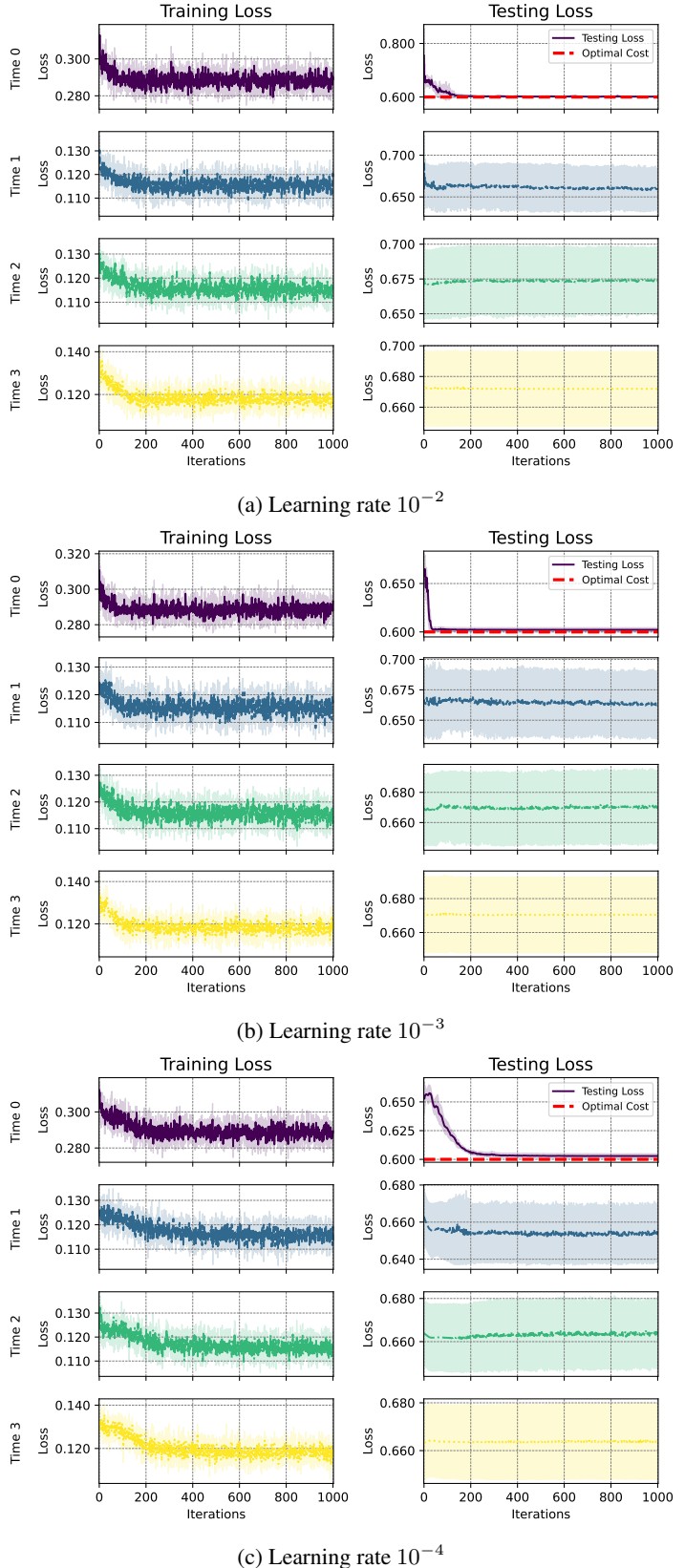

(a) Learning rate $10^{-2}$

(b) Learning rate $10^{-3}$

(c) Learning rate $10^{-4}$

Figure 30: Example 1: Sweep of learning rates. DPP results, asynchronous stopping.

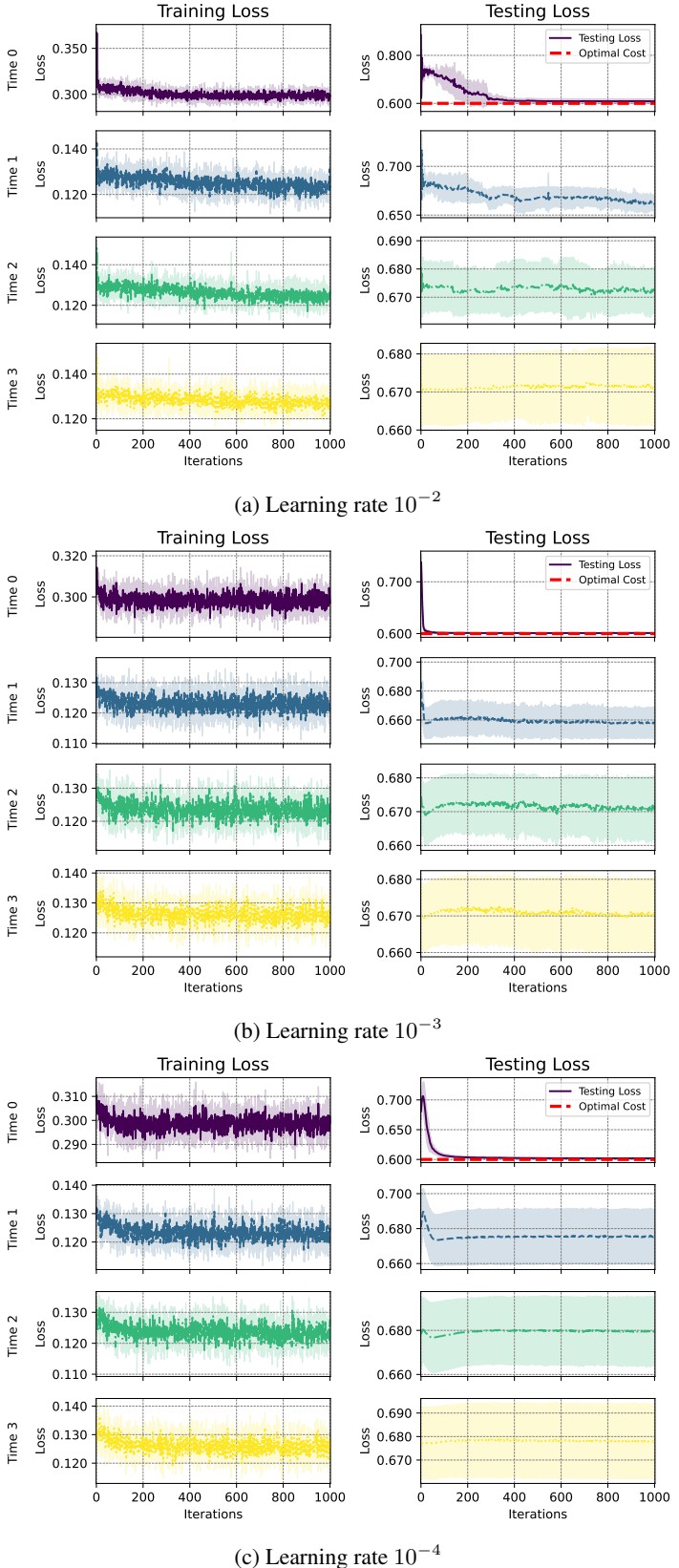

(a) Learning rate $10^{-2}$

(b) Learning rate $10^{-3}$

(c) Learning rate $10^{-4}$

Figure 31: Example 1: Sweep of learning rates. DPP results, synchronous stopping.

