# OpenReview forum: "Learning to Stop: Deep Learning for Mean Field Optimal Stopping"
_ICML.cc/2025/Conference — ICML 2025 poster_

### Official Review · Reviewer_DLNT · 2025-03-12

**Overall Recommendation:** 4

**Summary:**

The authors consider mean field control problems where agents must perform optimal stopping, i.e., taking an action that stops the evolution of their state. The cooperative optimal stopping problem is theoretically reduced into a standard mean field control problem by extending the state with information of whether an agent already stopped or not, similar to prior work in the continuous setting. Authors then propose direct learning-based solution algorithms minimizing the costs, which are appropriate due to the dimensionality of the problem (space of mean fields is continuous).

## update after rebuttal
I thank the authors for their detailed response. My questions have been addressed and the clarifications have been helpful. I see no reason to reject the work, and increased my score.

**Claims And Evidence:**

The theoretical claims in the submission are supported rigorously by proofs. The algorithms are supported by experiments and qualitative analysis in the main text. Additional details were printed in the Appendix.

**Essential References Not Discussed:**

N/A

**Experimental Designs Or Analyses:**

The experiments contain an extensive verification on example problems, covering the entire set of possibilities in the model. They look good to me, in particular Figs. 1-7. Some of the example problems are somewhat artificial. It may be worth mentioning that the common noise in experiment 6 is an empirical extension beyond the theoretical framework, but I understand it should not be a problem to extend the framework accordingly.

**Methods And Evaluation Criteria:**

The proposed algorithms use neural network learning methods by minimizing the loss directly, which makes sense given the dimensionality of the problem (space of mean fields is continuous). The new benchmark datasets make sense for the paper as well, as it considers a new special case of mean field control problems.

**Other Comments Or Suggestions:**

Typo line 788 "folliwng"

**Other Strengths And Weaknesses:**

- The exposition and clarity of the paper is good, as is the experimental verification and theoretical rigor.
- The originality of the theoretical results is slightly limited, as mean field optimal stopping problems and the reduction to MFC have been proposed in prior work.
- The significance is slightly limited, as the paper deals with a subset of two types of problems, though nevertheless both optimal stopping and mean field control problems in themselves are still sufficiently general.

**Questions For Authors:**

I may have missed it, but is it mentioned that agents must always stop before the time horizon T? Or is it allowed to never stop? If it is not mentioned, I think it is an important assumption.

**Relation To Broader Scientific Literature:**

The discussion with prior work in the areas of mean field control and continuous-time mean field stopping problems is sufficient. Moreover, references to prior work are made whenever appropriate in proofs or around theoretical claims.

**Theoretical Claims:**

The theoretical claims look largely correct to me (Thm. 3.2, 4.1, 4.2) and follow convergence rates and DPP results in the literature.

Minor question:
- In induction step of Lemma A.1, line 694, why is $\mathbb E[\lVert \nu^N_n - \nu_n \rVert] \leq (L_F (1 + L_P))^n \mathbb E[\lVert \nu^N_0 - \nu_0 \rVert]$? It looks like the induction assumption only specifies $\mathcal O(1 / \sqrt N)$ and since we have two terms, there should be an additional term from $|\mathcal S| / 4 / \sqrt N$?

---

> ### Author Rebuttal · Authors · 2025-04-01
>
> We sincerely thank the reviewer for the constructive suggestions to improve our paper. We provide detailed responses to each of your questions below.
>
> **(Theoretical Claims)** We would like to thank the reviewer for spotting a typo in the proof of Lemma             A.1. We corrected the proof by modifying lines 694, 695, and 747 in the original manuscript. Following the argument presented, in the end we obtain with the following inequality:
>
>   $E\left[\|\nu_{n+1}^{N,p} - \nu_{n+1}^{p}\|\right]\leq \frac{|S|}{4\sqrt{N}}(\frac{1 - K^{n+1}}{1- K}) + K^{n+1}E[\lVert \nu^{N,p}_0 - \nu^p_0 \rVert],$
>
> where $K=(L_F (1 + L_P))$. We remark that this updated inequality does not affect our conclusion as the convergence rate remains the order of $\mathcal O (1/\sqrt N)$.
>
>
>
> **(Experimental Designs or Analyses)** Although we agree that the experiments are relatively simple (compared with real-world applications), establishing a solid theoretical framework and  validating our algorithms through progressively more complex examples was a necessary step towards tackling real-world applications. We want to emphasize that all the experiments presented are intended as proof-of-concept demonstrations. Each of them was carefully designed to demonstrate specific features of MFOS.  As mentioned in lines 406-411, we include common noise in Example 6 to show that our method has potential beyond the theoretical framework covered in our proofs (see  Appendix E.6 for more details on this example). We will add more explanations about  how to extend the theory to cover the common noise.
>
>
> **(Originality and Significance)** Multi-agent problems have become a central topic in the machine learning community. The optimal stopping problem has not yet been extensively studied in the literature in a multi-agent scenario. MAOS fits neither in classical optimal stopping, nor in classical MDP theory.
> To the best of our knowledge, our work is the first to propose deep learning algorithms to efficiently address this class of problems, through the lens of mean field MDPs. If the reviewer has specific suggestions about potential improvements, we would be grateful and we would try to provide a more detailed answer.
>
> **(Question)** Thank you for this clarification question. In the setting we consider, all the agents are implicitly forced to stop when the problem reach the time horizon $T$. Concretely, we enforce $p_T^i(\cdot) = 1$. In this way, the stopping time $\tau^i$ defined at line 146 is always less than or equal to $T$. We will clarify this point in the text.

---

### Official Review · Reviewer_eVc6 · 2025-03-13

**Overall Recommendation:** 4

**Summary:**

The goal of this paper is to extend the classical optimal stopping problem to multi-agent systems. While existing theoretical results primarily address continuous-time settings—such as those arising in options pricing—this work specifically studies discrete-time and discrete-state scenarios. The central approach is based on a mean-field approximation, where the distribution of agents across states is tracked rather than modeling each agent’s environment individually. The authors prove that the mean-field approximation converges to the multi-agent solution with a quantifiable convergence rate and validate the effectiveness of their method experimentally.

**Claims And Evidence:**

The two central claims are:

1) The convergence of the mean-field approximations
2) The DP solution to the mean-field approximation

To the best of my knowledge, the proofs for both statements appear correct and support by evidence.

**Essential References Not Discussed:**

N/A

**Experimental Designs Or Analyses:**

The experiments presented in the paper supports the core theoretical results that mean-field approximation can solve multi-agent stopping problem at least in the small scale (small state dimension).

**Methods And Evaluation Criteria:**

The focus of the paper is primarily theoretical with simple experimental setup demonstrating the functioning behaviour of the proposed method.

**Other Comments Or Suggestions:**

l78 "time yields to an approximate optimal" -> " an approximate optimal"
l89 "relay on the interpretation of MFOS" -> "rely on the interpretation of MFOS"
l117 "then she is incurred the cost:" -> "then it incurs the cost"
l788 "folliwng" -> "following"

**Other Strengths And Weaknesses:**

N/A

**Questions For Authors:**

N/A

**Relation To Broader Scientific Literature:**

I would be curious if the authors could expand on the connections between their work and the options framework in reinforcement learning. They have a similar concept of termination / cost switching function and it would be worth drawing the parallel between both. For example "When Waiting Is Not an Option: Learning Options with a Deliberation Cost". The option framework effectively models when it is valuable to "stop" the current policy and switch to another one by comparing the value of both decisions. This seems to mirror the paradigm presented in this paper.

**Theoretical Claims:**

I reviewed theorem 3.2 (A.2 and A.3) which appears to be correct.

---

> ### Author Rebuttal · Authors · 2025-04-01
>
> We sincerely thank the reviewer for the helpful suggestions to improve our work, especially for raising the connection to the options framework in reinforcement learning. While we agree that this seems connected to MFOS in spirit, we believe that we cannot make one fit in the other. We list specific points below.
> - **Options framework**: There are three main challenges that make it difficult to compare our work directly with the options framework.
>     1. Once an agent decides to stop, they stay in their current state until terminal time $T$, whereas in the options framework, when the current option terminates, the agent has the possibility to pick a new option and her state keeps changing.
>     2. The "termination" criterion in our framework is controlled by the agent itself whereas in the options framework, the termination of the option is exogeneous to the agent and most of the time completely random.
>     3. We are working in a finite-horizon setting with non-stationary policies whereas the options framework seems particularly relevant for infinite horizon discounted setting;
> - **SMDP**: Our problem’s non‐Markovian nature differs from the semi‐MDP formulation found in options theory. In a semi‐MDP, the next state depends only on the current state and action, while the transition time can be random. In contrast, in our original framework, the transition depends on the agent’s entire history: the dynamics depends on whether the agent has stopped before or not. To make the problem markovian again, we need to introduce the augmented state space (Section 2.3) to capture the status of stopping decisions.
>
> Given these limitations, we believe that it is not possible to represent mean-field optimal stopping as an option framework, although the two share some similarity in spirit. We are open to further discussions.
>
> In the final version, we will include a summary of the above discussion on the options framework in RL. The typos will be corrected in the final version.
> We hope these arguments clarify our approach and may improve the reviewer’s assessment. Of course, we remain open to further discussion on these points.

---

### Official Review · Reviewer_F1p8 · 2025-03-16

**Overall Recommendation:** 4

**Summary:**

The paper introduces a mean field optimal stopping (MFOS) framework as an approximation to multi-agent optimal stopping (MAOS) problems where many agents must decide when to stop a stochastic process. The authors establish a theoretical foundation for MFOS, proving that solving the infinite-agent (mean field) version is a good proxy for the finite-agent problem. In particular, they show that the MFOS solution yields an optimal stopping policy for the $N$-agent case with an $O(1/\sqrt{N})$ optimality gap. They also derive a dynamic programming principle (DPP) for MFOS, analogous to Bellman’s principle in classical optimal stopping. Algorithmically, the paper proposes two deep learning methods to compute near-optimal stopping policies: (i) a Direct Approach (DA) that learns the stopping decision by forward-simulating full trajectories and training a neural network to decide stop/continue at each time, and (ii) a Dynamic Programming (DP) Approach that leverages the DPP by training neural networks backward in time (from the horizon to start) to approximate the value function and optimal stopping rule. Both methods output a time-dependent stopping probability for each state of an agent, effectively learning a randomized stopping policy that depends on the entire population distribution. The paper demonstrates these methods on six benchmark scenarios of increasing complexity. The experiments show that the learned policies closely achieve the intended objectives and scale to state spaces with up to 100 states (making the mean-field state distribution a 300-dimensional input). This work appears to be the first to formally define and solve MFOS in discrete time with finite state spaces, providing both new theory and scalable computational tools.

**Claims And Evidence:**

Overall, the paper’s claims are credible and backed by clear evidence, with no glaring over-claims.
The paper’s main claims are well-supported by both theoretical arguments and empirical evidence. First, the authors claim that the mean-field model provides a good approximation for the original $N$-agent problem. This is backed by a rigorous theorem (Theorem 3.2) establishing an $ε$-approximation: the optimal cost achieved by the mean-field solution is within $O(N^{-1/2})$ of the true $N$-agent optimum. They not only prove this result under appropriate assumptions, but also corroborate it with a numerical experiment: applying the learned MFOS policy to finite-$N$ simulations shows that the optimality gap and distribution error shrink on the order of $N^{-1/2}$, consistent with the theorem. In Figure 2, for example, the L2-distance between the empirical $N$-agent distribution and the mean-field distribution, as well as the cost suboptimality, clearly decays at the predicted $\sim N^{-1/2}$ rate. Secondly, the claim of a valid dynamic programming principle for MFOS is supported by a formal derivation (Theorem 4.1) and reference to known results in mean-field control theory. The authors provide a proof outline (with full details in the appendix) showing the DPP holds for their mean-field state-value function, thus substantiating this theoretical claim. Thirdly, the paper claims that the proposed deep learning algorithms can effectively learn optimal stopping rules and handle high-dimensional problems. This is evidenced by multiple experiments: for simpler cases where the true optimum is known, the training loss converges to the provably optimal value and the learned stopping decisions match the expected optimal behavior (e.g. in the “Towards the Uniform” example, the loss converges to the analytically computed optimal value and the population indeed spreads to a uniform distribution). In more complex scenarios without closed-form solutions, the authors analyze the learned policies and outcomes to show they make intuitive sense (e.g. under congestion, agents slow down stopping; under asynchronous vs synchronous stopping, the costs differ as expected). All major claims – from theoretical approximation to the success of the algorithms – are supported by either proofs or convincing experimental results. We did not identify any broad claims that lacked support. If anything, the assertion that this approach “opens new directions for scalable MAOS” is forward-looking but reasonable given the demonstrated scalability.

**Essential References Not Discussed:**

While the paper covers most of the recent and theoretical literature relevant to MFOS, there are a couple of classical works in optimal stopping that were not mentioned and perhaps could have been cited for completeness. In particular, the Least-Squares Monte Carlo (LSM) method by Longstaff & Schwartz (2001), a seminal technique for approximating optimal stopping policies (especially in the context of pricing American options), is not referenced. LSM is a widely known baseline for high-dimensional single-agent optimal stopping problems and could provide context on how practitioners traditionally handle large state spaces (via simulation and regression). Including it would have highlighted the differences: LSM uses basis function regression and does not naturally extend to multi-agent interactions, whereas the proposed approach uses neural networks and mean-field coupling. Similarly, the approximate dynamic programming approach by Tsitsiklis & Van Roy (1999) for optimal stopping is an important early work in the machine learning literature that demonstrated how to approximate the value function for stopping problems in high dimensions. This work was not cited either. While these omissions do not detract from the paper’s contributions, mentioning them could have strengthened the background.

**Experimental Designs Or Analyses:**

I found the experimental design to be well thought-out and the analysis of results to be sound and credible. Each experiment’s outcome is interpreted with respect to the theory (e.g., checking if mean-field and multi-agent results agree, or if costs decrease as expected), which strengthens the validity of the conclusions.

**Methods And Evaluation Criteria:**

The methods chosen are very well suited to the MFOS problem setting, and the evaluation — using custom but relevant scenarios rather than off-the-shelf datasets — is thorough and appropriate for demonstrating effectiveness on this problem.

**Other Comments Or Suggestions:**

The paper is generally well-presented, but we have a few minor comments and suggestions that could improve it further:
- Typos: There are a handful of minor grammatical issues. For example, in the introduction the authors state “We will refer to this setting at multi-agent optimal stopping (MAOS)”, which should read “refer to this setting as multi-agent optimal stopping.” Another instance is in the Contributions section: “Our theoretical results relay on the interpretation of MFOS problems as MFC problems” – here “relay” should be “rely”. The same sentence ends with “opens up new direction to study MFOS problems,” where it should likely be “opens up new directions.” These are very minor and do not affect understanding, but fixing them would improve the polish of the text.
-	Highlight Randomization Requirement: The need for randomized stopping policies is an important insight of this work (and of mean-field decision processes in general). The appendix example (Appendix A.1) illustrates it excellently. It might be worth alluding to this point in the main text as well, perhaps in Section 2.1 (Motivation and challenges) when discussing why single-agent methods cannot be applied. The authors do mention that in multi-agent setting “we allow agents to stop at different times” and that treating the whole system as one agent is flawed, which is related. But an explicit statement like “(Unlike single-agent stopping, an optimal policy in the multi-agent case may require randomizing: e.g., having 30% of agents stop now and others later.)” would prepare the reader for the introduction of $p_n(x) \in [0,1]$ as a control. This is a minor suggestion for pedagogical clarity.
-	Include Classical Baselines: As noted, the paper omits references to classical optimal stopping solution methods (LSM, etc.). While those methods can’t handle the multi-agent aspect, it could be instructive to compare at least in the single-agent limiting case. For instance, for Example 1 (which is essentially a single-agent OS with an extra population cost term), one could compute the solution via a backward dynamic programming (since state space is small) or LSM and show the neural network achieves the same result. The authors did something similar by analytical calculation, which is great. If space permits in a final version, a brief mention that classical methods (like regression-based DP) would struggle as state dimension grows or with the need to incorporate distributions could strengthen the argument for why a new deep learning approach is warranted.
-	Experimental Details: The appendix provides a lot of details on architecture and hyperparameters. We think it might be useful to mention in the main text (Section 6) a summary of the network architecture used for the experiments. For example, one or two sentences like “In all experiments, we use feed-forward neural networks taking the state (and distribution) as input; for asynchronous stopping the network input is [state, distribution] and for synchronous it is just [distribution], with appropriate embedding layers for state. We train using Adam with learning rate X, etc., as detailed in Appendix E.” This would give readers a concrete sense of the model complexity. Currently, one has to jump to Appx. E or C to find that information. Again, this is a minor suggestion for completeness.

**Other Strengths And Weaknesses:**

Strengths:
- Innovative Problem Formulation: This work is the first to formalize MFOS in a tractable discrete setup and solve it computationally. This is a significant theoretical step – bridging multi-agent optimal stopping and mean-field control – that opens up a new line of research. The conceptual connection made between MFOS and mean-field control (MFC) is novel and non-trivial, providing a fresh perspective on stopping problems by leveraging control theory tools.
- Combination of Theory and Practice: The paper excels in providing both theoretical guarantees and practical algorithms. It’s commendable that the authors prove an approximation rate for the multi-agent problem and derive a Bellman-like principle, and then use those insights to design algorithms. This synergy of theory and deep learning is a strong point – it lends credibility to the methods and also guides their design (e.g., the DP approach directly follows from the DPP theorem).
-	Scalability and Demonstrated Performance: The proposed methods show the ability to handle high-dimensional state distributions (hundreds of dimensions) and complex scenarios. Solving an optimal stopping problem in a 100-state environment (with a 300-dimensional input when including the distribution) is non-trivial, and the paper demonstrates this feat. The fact that they can train neural networks on distribution inputs and still converge to near-optimal policies suggests the approach is robust and scalable. This is a crucial strength since one major goal was to enable solving very large multi-agent problems that are otherwise infeasible.
-	Comprehensive Experiments and Insights: The range of experiments is a strength in itself. By examining six different examples, the authors validate their method under various conditions and also extract interesting insights (e.g., showing how asynchronous stopping outperforms synchronous stopping in terms of cost, which provides guidance for practitioners on what type of stopping rule to allow). They also examine the trade-offs between the two algorithms (DA vs DP), noting memory vs speed considerations, which adds depth to the evaluation. The paper doesn’t treat the method as a black box; it investigates why and when each approach works better, which improves the clarity and usefulness of the contribution.

Weaknesses:
- Restricted to Finite State Mean-Field: A notable limitation is that the approach is confined to finite state spaces (discrete state distributions). The authors themselves acknowledge that continuous-state MFOS leads to an infinite-dimensional problem which is intractable. While focusing on a finite state approximation is a reasonable and necessary step, it means the method might require state discretization for truly continuous problems (e.g. stopping problems in $\mathbb{R}^d$), which could be a source of approximation error or computational burden if the state must be finely discretized.
- Computational Cost: The paper does not deeply discuss the runtime or sample complexity of the methods. Training deep networks for each new MFOS problem could be computationally intensive, especially as the state space or time horizon grows. The authors do highlight the memory issue and how DP circumvents it, but there is little quantitative data on training times or how performance scales with problem size. Thus, a practical weakness is that the method might be computationally heavy for extremely large-scale problems (though still far better than brute force on the $N$-agent problem, of course).

**Questions For Authors:**

What are the computational requirements of your approach as the problem size grows? In particular, how does training time scale with the number of states or the time horizon $T$? For example, if we were to double the state space size or consider a horizon $T=10$ instead of $T=4$, would the approach still be feasible? Any information on the runtime or memory usage for your largest experiment (Example 6 with a 100-state grid and presumably a larger $T$) would help assess the practical scalability of the method.

**Relation To Broader Scientific Literature:**

The paper demonstrates a strong grasp of the relevant literature: it bridges the gap between single-agent optimal stopping methods and multi-agent mean field methods. The key contributions – establishing the MFOS approximation and solving it via deep learning – are clearly contrasted with prior works: none before have provided a computational solution for MFOS in discrete time. This positions the paper as a novel contribution that extends known ideas (optimal stopping, mean field control) into a new combined realm. I did not identify any major prior study that was directly relevant and omitted, aside from some classical methods discussed below. Overall, the authors’ related work section and citations indicate a high level of familiarity with the broader scientific context.

**Theoretical Claims:**

I reviewed the provided proofs for correctness and clarity, focusing on Theorem 3.2 and Theorem 4.1, 4.2, and both were convincing and free of apparent error.

---

> ### Author Rebuttal · Authors · 2025-04-01
>
> First of all, we would like to sincerely thank the reviewer for their detailed review, which highlights our contributions and originality, and for the constructive suggestions to improve our manuscript.
>
> **(Restricted to Finite State Mean-Field)**. We agree with the reviewer that focusing on a finite‐state mean‐field setting is a limitation, and we plan to explore continuous settings in future work. However, we would like to emphasize that this is the first work to both model and propose deep learning algorithms for multi‐agent optimal stopping in this setting. We believe this represents a fundamental step before tackling more complex scenarios.
>
> **(Typos)**. We thank the reviewer for taking the time to carefully read our work. We have corrected the typos and improved the readability in the final version.
>
> **(Highlight Randomization Requirement)**. The randomized stopping policies are a key aspect of this work, as emphasized by the reviewer. Following the proposed modifications, we have strengthened the argument in the main text.
>
> **(Essential References Not Discussed and Classical Baselines).** We thank the reviewer for directing us to these important works on optimal stopping. In response, we will update the introduction to more clearly position our method relative to LSM and the work of Tsitsiklis & Van Roy (1999), comparing the different approaches. Although our first two examples already compare our algorithm’s solution against the analytical solution obtained via dynamic programming, we agree that conducting additional comparisons with established classical methods will strengthen our argument.
>
> **(Experimental Details)**. We agree with the reviewer that including a description of the network´s architecture in the main text (section 6) will help the reader better understand the results. We thank the reviewer for the valuable suggestions, which we will incorporate into the final version.
>
> **Runtime and Memory Usage** -
> Regarding the reviewer's question and observations on runtime and memory usage we have briefly commented on the training time and computational resources in Appendix D. To provide a more ***quantitiative analysis***, we list the memory usage and required training time per 100 iterations in the following table, for a varying size of problem with different time horizons and dimensions. We fix the batch size to be 128 and deep networks to have around 260k parameters.
>
> - From the table, we see that **memory usage** for the direct approach (DA) scales as $O(DT)$, where $D$ is the problem dimension and $T$ is the time horizon, while the dynamic programming approach (DP) only requires memory of $O(D)$. We also observe that DA in general requires more memory than DP, which makes DP the most preferable approach for problems with long time horizons or very high dimensions, as we have discussed in the paper.
>
> - As for the **running time**, the time horizon $T$ has a crucial impact on the required time per $100$ iterations, while the dimension plays a relatively minimal role. While DA tends to run slower per hundred iterations, we want to stress that in practice it takes less than $2000$ iterations for DA training, but would usually requires around $200 * T$ total iterations for DP training. Therefore, DA still could be faster in training time compared with DP when the memory usage is affordable.
>
> Based on these quantitative data, we believe that our proposed approaches are quite scalable, and therefore still feasible for long-horizon, high dimensional problems and are of pratical impact in real scenarios.
> | Memory| T=10 dim=100|T=30 dim=100|T=50 dim=25|T=50 dim=64|T=50 dim=100|
> |:----------:|:----------:|:----------:|:----------:|:----------:|:----------:|
> |Direct Approach|3.8GB|10.0GB| 4.7GB | 10.1GB |16.1GB|
> | Dynamic Programing| 0.9GB | 0.9GB | 0.7GB | 0.8GB|0.9GB |
>
> | Time/100 iter| T=10 dim=100|T=30 dim=100|T=50 dim=25|T=50 dim=64|T=50 dim=100|
> |:----------:|:----------:|:----------:|:----------:|:----------:|:----------:|
> |Direct Approach| 14s | 42s | 33s | 56s | 71s|
> | Dynamic Programing| 2s | 4s | 6s | 6s |  6s |

---

### Decision · Program_Chairs · 2025-05-01

**Decision:**

Accept (poster)

**Comment:**

This work deals with Multi-Agent Optimal Stopping (MAOS) problem, in discrete time and space, leveraging Mean Field Optimal Stopping (MFOS) as an approximation of MAOS when the number of agents is large. In this setting, authors  1) show that MFOS is indeed a good approximation of MAOS, providing a convergence rate of order $1/\sqrt{N}$ where $N$ is the number of agents 2) Derive a dynamic programming principle for MFOS 3) propose two deep-learning approach to solve MFOS, and demonstrate their performances with numerical experiments.

All reviewers acknowledge the relevance of the approach and the soundness of the theoretical results. The fact that results are supported by theoretical statements and empirical evidence is also much appreciated.

For this reason, I recommend acceptance.